# Observations of OH-airglow from ground, aircraft, and satellite: investigation of wave-like structures before a minor stratospheric warming

Sabine Wüst[1*], Carsten Schmidt[1], Patrick Hannawald[2], Michael Bittner[1,2], Martin G. Mlynczak[3], James M. Russell III[4]

[1] Deutsches Fernerkundungsdatenzentrum, Deutsches Zentrum für Luft- und Raumfahrt, , 82234 Oberpfaffenhofen, Germany
[2] Institut für Physik, Universität Augsburg, Augsburg, Germany
[3] NASA Langley Research Center, Hampton, USA
[4] Center for Atmospheric Sciences, Hampton, USA

*Correspondence to*: Sabine Wüst (sabine.wuest@dlr.de)

**Abstract**

In January and February 2016, the OH-airglow camera system FAIM (Fast Airglow Imager) measured during six flights on board the research aircraft FALCON in Northern Scandinavia. Flight 1 (14[th] January 2016) covering the same ground track in several flight legs and flight 5 (28[th] January 2016) along the shoreline of Norway are discussed in detail in this study. The images of the OH-airglow intensity are analysed with a two-dimensional FFT regarding horizontal periodic structures between 3 km and 26 km horizontal wavelength and their direction of propagation. Two ground-based spectrometers (GRIPS, Ground based Infrared P-branch Spectrometer) provided OH-airglow temperatures. One was placed at ALOMAR, Northern Norway (Arctic Lidar Observatory for Middle Atmosphere Research; 69.28° N, 16.01° E) and the other one at Kiruna, Northern Sweden (67.86° N, 20.24° E). Especially during the last third of January 2016, the weather conditions at Kiruna were good enough for the computation of nightly means of gravity wave potential energy density. Coincident TIMED-SABER (Thermosphere Ionosphere Mesosphere Energetics Dynamics, Sounding of the Atmosphere using Broadband Emission Radiometry) measurements complete the data set. They allow for the derivation of information about the Brunt-Väisälä frequency and about the height of the OH-airglow layer as well as its thickness.

The data are analysed with respect to the temporal and spatial evolution of mesopause gravity wave activity just before a minor stratospheric warming at the end of January 2016. Wave events with periods longer (shorter) than 60 min might mainly be generated in the troposphere (at or above the height of the stratospheric jet). Special emphasis is placed on small-scale signatures, i.e. on ripples, which may be signatures of local instability and which may be related to a step in a wave breaking process. The most mountainous regions are characterized by the highest occurrence rate of wave-like structures in both flights.

# 1    Introduction

The results presented here are part of the international initiative ROSMIC (Role Of the Sun and the Middle atmosphere/ thermosphere/ionosphere In Climate) and the German program ROMIC (Role Of the Middle atmosphere In Climate). One goal of ROMIC was to investigate coupling mechanisms which connect atmospheric layers from the ground up to the top of the middle atmosphere and vice versa. The project GW-LCYCLE, which was part of ROMIC, addressed questions concerning the life cycle of gravity waves, i.e. their excitation, propagation, and dissipation.

During the field campaign in winter 2015/16 in Northern Scandinavia, ground-based as well as airborne airglow measurements were conducted. At the ground, two spectrometers (GRIPS, Ground based infrared P-branch spectrometer, one at Kiruna, 67.86° N, 20.24° E, and one at ALOMAR, Arctic Lidar Observatory for Middle Atmosphere Research; 69.28° N, 16.01° E) and one camera (FAIM, Fast Airglow Imager, at Kiruna) were operated. An additional FAIM system with small aperture was mounted on the DLR (Deutsches Zentrum für Luft- und Raumfahrt) research airplane FALCON. Six flights were conducted in January and February 2016 in Northern Scandinavia. The airglow observations refer to the height range of ca. 80–90 km, all other measurements on board the FALCON address heights of ca. 20 km and below. Flight 1 (14[th] January 2016) covering the same ground track in several flight legs and flight 5 (28[th] January 2016) with a long flight leg almost parallel to the shoreline of Norway allow for the discussion of different wave activity features and were therefore chosen for a detailed discussion in this study.

Airglow camera measurements on board a research aircraft and therefore covering a wider spatial range are very rare. To our knowledge the only other system with a very good spatial and temporal resolution flown on an aircraft is the one described in Pautet et al. (2016). In contrast to the airglow imaging system (AMTM, Advanced Mesosphere Temperature Mapper) used by those authors during the DEEPWAVE campaign above New Zealand, the airborne FAIM does not allow for the derivation of OH rotational temperatures. FAIM is based on a InGaAs 320 px $\times$ 256 px sensor and integrates over wavelengths from 0.9 µm to 1.65 µm with a temporal resolution of 1 s. It was optimized for the study of small-scale features (some 100 m to some 10 km depending on the zenith angle and the optics used) in airglow intensity and therefore it has a significantly smaller aperture (in this case 27.3° $\times$ 33.9°) and covers a wider spectral range resulting in a higher spatial and temporal resolution at the sacrifice of geographical coverage (Hannawald et al., 2016). While Pautet et al. (2016) analysed a specific gravity wave event, we concentrate here on the temporal and spatial development of periodic structures in the range of ripples and bands during the two flights mentioned above (section 4.2.2 and 5). Due to this unprecedented spatial and temporal resolution, the focus of this manuscript is especially on the question whether the activity of these small-scale features is enhanced above possible tropospheric gravity wave sources. Furthermore, their activity is studied under different meteorological conditions.

Information about the temporal development of the potential energy density of larger-scale gravity waves are derived based on the GRIPS measurements at Kiruna (section 4.2.1 and 5). Due to varying weather conditions the temporal resolution of

the time series is best during the last third of January 2016. Unfortunately, the weather conditions did not allow these analyses during the same time period for the ALOMAR GRIPS data.

For the calculation of the density of wave potential energy, we compute the (angular) Brunt-Väisälä (BV) frequency based on coincident TIMED-SABER (Thermosphere Ionosphere Mesosphere Energetics Dynamics, Sounding of the Atmosphere using Broadband Emission Radiometry) temperature and OH-B channel volume emission rate (VER) measurements. We use the latter also in order to learn more about the OH-layer height and thickness (section 4.1 and 5).

All results are interpreted in the context of the minor sudden stratospheric warming (SSW) which happened at the end of January 2016 (Dörnbrack et al., 2018). As SSW events are associated with dynamical changes in the stratosphere and mesosphere over several days, effects on gravity waves in the upper mesosphere / lower thermosphere can be expected and have also already been observed and/or modelled: Yigit and Medvedev (2012), for example, report that GW activity increases by a factor of 3 in the course of the warming modelled by them. Liu (2017) point out that at high latitude in the winter hemisphere the momentum flux varies rapidly during the SSW. Afterwards, the magnitude of the mesospheric momentum flux decreases significantly. His findings agree with the observations of GW momentum flux changes during a SSW published by Wright et al. (2010), France et al. (2012), Thurairajah et al. (2014) and Ern et al. (2016), for example. Liu (2017) argues that the rather rapid change of the winter jet system is expected to be a source of GW variability during SSW as GWs can be excited by imbalance of jet flow (O'Sullivan and Dunkerton 1995; Zhang 2004).

## 2    Data

### 2.1    GRIPS

During winter 2015/16, ground-based airglow observations were carried out with the infrared spectrometers GRIPS 9 at Kiruna (67.86° N, 20.24° E), Sweden and GRIPS 14 at ALOMAR (Arctic Lidar Observatory for Middle Atmosphere Research; 69.28° N, 16.01° E), Norway.

GRIPS instruments are based on a monochromator with 163 mm focal length (Czerny-Turner setup with crossed beam configuration) and a thermoelectrically cooled 512 px InGaAs photodiode array. They observe the spectral range between 1.5 μm and 1.6 μm, which includes OH(3-1) and OH(4-2) vibrational transitions (OH(3-1) Q- and P-branches, OH(4-2) R- and Q-branches up to the first line of the OH(4-2) P-branch). The spectral resolution is ca. 3.1 nm at a wavelength of 1550 nm. The field of view (FoV), over which the instruments integrate, is mainly governed by the F-number of the polychromator (#F3.6) because the instrument is operated with no further objective lenses. In standard setup, the temporal resolution is 15 s. Details about the instrument are provided in Schmidt et al. (2013).

Rotational temperatures are derived operationally from the first three $P_1$ lines of the OH(3-1) transition, $P_1(2)$, $P_1(3)$, and $P_1(4)$. Under the assumption of local thermodynamic equilibrium, the intensity of these lines follows a Boltzmann distribution. The only variable on which this distribution depends is the temperature. Therefore, the relation of the intensity

of these lines allows calculating the rotational temperature, Einstein coefficients and term values of the respective rotational level provided (Meinel, 1950, Krassovsky et al., 1962, Mies, 1974, Schmidt et al., 2013).

In this study, one minute mean and nightly mean temperature are used. One minute mean values typically have a precision of up to $\pm 8$ K (from error propagation calculation of the 15s values), but the exact value strongly depends on the emission intensity, which can be highly variable. Due to the high number of measurements, the error of nightly mean values is much lower. For the temperature derivation, the measured spectrum between 1.5 µm and 1.6 µm is approximated using a low pass filter. The uncertainty is calculated from the standard deviation of the residuals. It is influenced by the observations conditions and by the instrument itself (readout noise of the detector, which is assumed to be constant, the photon or shot noise and the dark current noise, which both refer to statistical variations and scale with the square root of the signal level and dark current level). The average pattern of the dark current noise along the photodiode array as well as a constant noise value are retrieved and subtracted. The major sources of uncertainty are therefore bad weather conditions (background intensity is increased by modulated by $H_2O$ absorption in the lower atmosphere), which disturb the noise reduction or make it impossible. Details about the temperature retrieval and the noise reduction are also provided in Schmidt et al. (2013). Airglow observations are only performed during darkness. Dense cloud coverage poses an obstacle for the measurements, thin clouds or fog increase the error.

At Kiruna, GRIPS 9 observed the airglow layer at a fixed zenith angle of 0°. Its FoV was approximately 25 km x 25 km. GRIPS 14 at ALOMAR was operated in a scanning mode (the FoV is changed four times within one minute then starting with the first FoV again, azimuth and zenith angles: (120°, 30°), (0°, 30°), (240°, 30°), (not available, 0°)) with one FoV in zenith direction. Concerning the size of the FoV, the time series referring to zenith direction at ALOMAR is comparable to the measurements at Kiruna. For more information about the GRIPS system, operated in standard and in scanning mode, see Schmidt et al. (2013), Wachter et al. (2015), and Wüst et al. (2018).

Observations at ALOMAR were performed between December 5, 2015 and February 3, 2016. Observations at Kiruna were carried out between January 14 and February 2, 2016. During the latter time period, the weather at ALOMAR was very variable. Information about gravity wave potential energy density (GWPED) according to Wüst et al. (2016) and the data quality criteria given therein were therefore not derived for ALOMAR. Nightly-mean temperatures, however, were successfully retrieved for 56 out of 61 nights at ALOMAR and for 19 out of 19 nights at Kiruna.

## 2.2    FAIM

Two-dimensional airglow observations were performed by FAIM (Fast Airglow Imager, for details about the ground-based instrument see Hannawald et al., 2016) on board the DLR aircraft FALCON.

The spectral range, over which the instrument integrates every second, is 0.9 µm–1.65 µm, the size of the InGaAs sensor is 320 px $\times$ 256 px (model "Xeva" manufactured by Xenics nv). The intensities recorded by FAIM include different airglow emissions (oxygen and hydroxyl), however, the influence of OH airglow dominates. The instrument is equipped with a three-

stage thermoelectric cooler. Due to limited available space, the instrument was mounted on the headmost aperture plate position of the aircraft. The observations were performed in near zenith direction (roll angle of the instrument w.r.t. the aircraft plane: -5°). The aperture angle of the optics was mainly limited by the small diameter window (approx. 70 mm), so a lens system with opening angles of 27.3° × 33.9° was used (larger angles resulted in stronger vignetting). In order to

maximise the geographical coverage, the camera was mounted with a yaw angle of -45°, making the image diagonally oriented to the flight track (compare Fig. 7 and 10). The spatial resolution was approximately 167 meters per pixel, the FoV covered an area of 43 km × 55 km. The exact values depend on roll (25° at maximum) and pitch (0°–5°) angles as well as on the height of the aircraft. The high temporal and spatial resolution allows especially the investigation of gravity waves with short wavelengths and short periods along the flight track.

During the GW-LCYCLE campaign, six night-time flights were conducted above Northern Scandinavia. At least large parts of the flights took place above the tropospheric cloud level. The camera delivered data for all six flights. Flight 1 (14[th] January 2016) and flight 5 (28[th] January 2016) were least disturbed by aurora or moon light. Therefore, they are subject of this study.

Additionally, an all-sky FAIM system was operated at Kiruna from January, 14[th] 2016 to February, 2[nd] 2016. Due to low

level clouds or fog at the times of the aircraft overpasses as well as some water vapour condensation occurring on the entrance optics during the very cold nights (-40°C), only few nights of this ground-based imager can be analysed. Therefore, these measurements are not part of this publication.

**2.3    TIMED-SABER**

The TIMED satellite was launched on 7 December 2001 and the on-board limb-sounder SABER delivers vertical profiles of kinetic temperature on a routine base from approximately 10 km to more than 110 km altitude with a vertical resolution of about 2 km until today (vertical sampling 300–400 m). The high vertical resolution is suitable for the investigation of gravity wave activity. About 1200 temperature profiles are available per day. The latitudinal coverage on a given day extends from

about 53° latitude in one hemisphere to 83° in the other. Due to 180° yaw manoeuvres of the TIMED satellite this viewing geometry alternates once every 60 days (Mertens et al., 2004; Mlynczak, 1997; Russell et al, 1999).

Kinetic temperatures are derived from the 15 μm $CO_2$ emissions. One of the main problems of deducing kinetic temperature values in the mesosphere and upper heights are non-LTE conditions (NLTE), i.e. conditions that depart from Local Thermodynamic Equilibrium. NLTE algorithms for kinetic temperature were employed in the SABER temperature retrieval

in version (v) 1.03 (Lopez-Puertas et al., 2004) as well as in v1.04 and 1.06 (Mertens et al., 2004, 2008). In v1.07, further improvements were made: $CO_2$ profiles from the Whole Atmosphere Community Climate Model (WACCM) were used in the retrieval algorithm in order to remove inconsistencies in the vertical structure of diurnal temperature tides (Remsberg et al., 2008), for example. A discussion of SABER v1.07 is provided by Remsberg et al. (2008) and García-Comas et al.

(2008). The WACCM results integrated in v1.07 were scaled to match a $CO_2$ trend model. V2.0 now uses $CO_2$ from an updated WACCM model. Furthermore, some reaction rates were changed in the $CO_2$ vibrational temperature model. Finally, this newest data version relies on recalibrated SABER radiances and on retrieved [O] values for all data (v1.07 used retrieved [O] for daytime measurements only and where the solar zenith angle < 85°). The information concerning v2.0 is taken from Dawkins et al. (2018).

Those authors also compared SABER temperature data v2.0 between 75 and 105 km with data from nine ground-based lidars deployed at different latitudes (Spitsbergen at 78.0°N, ALOMAR at 69.3°N, Kühlungsborn at 54.1°N, Boulder at 40.1°N, Fort Collins at 40.6°N, Logan at 41.7°N, Arecibo at 18.4°N, Cerro Pachon at 30.3°S, and McMurdo at 77.8°S). Also for this SABER version holds that the kinetic temperatures (more precisely the seasonal mean in this case) derived by the satellite and the validation instrument agree well. The smallest absolute temperature difference is found between 85 and 95 km height, where the respective SABER and lidar uncertainties were smallest.

We use TIMED-SABER temperature and OH-B channel data (volume emission rates, VER) in its latest version (2.0) within a square of 300 km edge length centred at ALOMAR and between 17 and 5 UTC for the derivation of additional information about the OH-airglow layer characteristics and about the (angular) Brunt-Väisälä (BV) frequency at mesopause height. These values are also taken for Kiruna, since both locations are not more than 300 km away from each other. For heights of 80–90 km, precision, systematic errors, and accuracy are specified with 1.8–3.6 K, 1.4–4.0 K, and 2.3–5.4 K for a single profile (according to Dawkins et al. (2018) who reproduced these figures from saber.gats-inc.com).

The OH-B channel covers the wavelength range from 1.56 to 1.72 μm, which includes mostly the OH(4-2) and OH(5-3) vibrational transition bands. The mean height difference between the OH(4-2)- and the OH(3-1)-emission, which is addressed by the OH*-spectrometers mentioned above, is approximately 500 m (von Savigny et al., 2012). Therefore, aspects derived from the OH-B channel concerning vertical movements, for example, also hold for the OH(3-1) layer.

The data was downloaded from the SABER homepage (saber.gats-inc.com).

## 3    Analysis

### 3.1    Derivation of wave potential energy density

From the GRIPS data, we derive the density of gravity wave potential energy $E_{pot}$ (GWPED) according to

$$E_{pot} = \frac{1}{2}\frac{g^2\overline{\widehat{T}'^2}}{N^2} \tag{1}$$

where

$g$ is the acceleration of gravity, ($g$=9.6 m/s² taking into account its height-dependence),

$N$ the (angular) BV frequency, and

$\widehat{T}' = T'/\widehat{T}$ the normalized temperature fluctuation. The overbar denotes the nightly average.

$E_{pot}$ is calculated for different period ranges. It is distinguished between periods shorter and longer than 60 min as they show different overall evolvement (concerning annual and semi-annual oscillations, for example). The extraction of the temperature fluctuations is based on the iterative calculation of sliding means. Since this approach results in a shortening of the smoothed data series, data of gaps of 20 min at most are interpolated and the time series is mirrored at the beginning and the end. Details can be found in Wüst et al. (2016).

For the derivation of the (angular) BV frequency $N$ vertically-resolved temperature profiles are needed:

$$N = \sqrt{\frac{g}{T}\left(\frac{dT}{dz} + \Gamma\right)} \tag{2}$$

where

$z$ is the height,

$T$ is the temperature and

$\Gamma$ the dry adiabatic lapse rate with  9.6 K/km.

Since GRIPS provides a temperature value which is vertically averaged over the OH*-layer, additional data are necessary for the calculation of $N$. As in Wüst et al. (2016, 2017a, 2017b), TIMED-SABER temperature information is used for this purpose. The OH*-equivalent (angular) BV frequency is calculated for the day of year (DoY) 1–60 of 2016 by weighting the

height-dependent squared BV-frequency values with the volume emission rate (VER) profiles. From time to time, the maximum of the VER is observed around 40 km in SABER profiles. Therefore, the calculation is restricted to the height range between 71 and 97 km (84 km ± 13 km, the height of 84 km corresponds to the mean height of maximum VER derived by Wüst et al. (2016) around ALOMAR based on SABER OH-B channel measurements for one year, in general slightly higher values are reported for other stations or other time periods, see Wüst et al. (2016, 2017a, 2017b) or Baker and

Stair (1988)). Since the SABER profile provides only a snapshot of the atmospheric situation, an error of 10% is assumed. This value includes the day-to-day variability (Wüst et al., 2017b).

## 3.2 Spectral analysis

In order to analyse the FAIM measurements, a two-dimensional Fast Fourier Transform (2D FFT) is applied. Sequences with high roll and pitch angles as they occur during turning manoeuvres were excluded since the size of the FoV and the spatial resolution change significantly within a short time. A steady shaking of the airplane limits the application of the 2D FFT: The shaking translates the FoV by several pixels in a quasi-periodic manner and applies a motion blur on the images. The translation does not allow deriving the change of the phase from consecutive images, but this information is crucial for calculating phase speed and period of the waves. The translation affects the whole image and therefore all wave crests within the image, the wavelength, which is derived for each image individually, is not influenced. The motion blur does not change the position of the wave crests, but it reduces the amplitude of the waves. The amplitude, however, is not used here. Only horizontal wavelength and the direction of propagation with a 180° ambiguity are computed.

The 2D FFT algorithm employed needs equidistant data. Therefore, the images are un-warped. As mentioned above, the camera is deployed at a roll angle w.r.t. the plane of -5° and at a yaw angle w.r.t. the plane of -45°. Therefore, two rotation matrices (one for the yaw axis and one for the roll axis) are used to convert the reference system of the instrument to the reference system of the airplane. The orientation of the airplane is also characterized by a roll, pitch, and yaw angle. Therefore, three rotation matrices are then applied to transform the reference system of the airplane to a world coordinate system (azimuth and elevation relative to the Earth's surface). The three required angles are taken from the flight metadata which are given with a temporal resolution of 1 s. The new pixel positions are then calculated by projecting the image in world coordinates to the airglow layer at 87 km height. Changes in the airglow layer altitude have only minor effects on the results (+/-5 km in the altitude layer corresponds to +/- 6% in the resolution and therefore also in the horizontal wavelength, calculated for a zenith angle of 5°). An additional flipping at the North-South axis brings the image to a satellite's view perspective. The scale is kept constant with 167 m/px (or 6 px/km) for all images allowing direct comparison of images at different times and angles. Before analysing the un-warped images, the stars in the images need to be removed since otherwise their signal may influence the 2D FFT spectra. This is done by applying a sliding median blur with a kernel of $17 \times 17$ px.

All images are reduced to 26 km × 26 km. This is the largest square size which does not contain any pixels outside the un-warped image region (marked in Fig. 1).

For each image, the mean is subtracted and a Kaiser-Bessel window ($\alpha=4$) is applied to let the borders of the area steadily decrease to zero. Zero-padding further optimises the calculation of the 2D-FFT.

After calculating the 2D FFT for each image (each flight consists of ca. 12,000 images), wavelength and angle of propagation are extracted for every significant peak in the spectrum (Monte-Carlo significance test with a significance level of 95%).

The algorithm is described in detail in Hannawald et al. (2019).

# 4    Results

## 4.1    Height, thickness and intensity of the OH-layer

During winter (DoY 1–60) 2002–2016, the averaged maximum of the volume emission rate, in the following denoted as OH*-layer height, and its averaged full-width a half maximum (FWHM) around ALOMAR vary between ca. 84.5–86.0 km

and 7.0–7.5 km, respectively, based on SABER data (see Fig. 2 (a) and (b) for a comparison with the mean over all years, and Fig. 3 (a) and (b) for details about the year 2016). Compared to the mean over all years, the FHWM for 2016 (thick blue line) can be characterized as low especially during the end of January and large parts of February. It shows a drop centred at DoY 43. The OH*-layer height is stable at 85.0–85.5 km until DoY 23 and rises by 2.5 km during the following ten days. Afterwards, it oscillates around ca. 87.5 km. Compared to the mean over all years, the OH-airglow layer altitude increases

and its width decreases from DoY 23 on.

Both, ground-based GRIPS and space borne SABER observations of the OH intensity, agree fairly well during the GW-LCYCLE campaign (see Fig. 4a). In the case of SABER, the peak intensity is used, which correlates very well with the intensity integrated over the analysed height range ($R^2$ of about 87%). The GRIPS instruments deliver only relative intensities. They are normalized to their respective mean. In particular, the intensities of GRIPS and SABER show

pronounced periodicities in the range of some days. At the end of January and the beginning of February, the intensities of all instruments reach their absolute minimum.

The same characteristics hold also for the temperature derived by the GRIPS instruments and the VER-weighted temperature calculated from SABER (see Fig. 4b). Over all, temperature and intensity show similar relative variations. The absolute temperature difference between GRIPS and SABER measurements referring to ALOMAR varies between ca. 0 K and 18 K.

This difference is not unusual for this height range, altitude and season when taking into account that the SABER and GRIPS measurements do not match exactly in place, time and addressed air volume (Wendt et al., 2013).

## 4.2    Periodic signatures

### 4.2.1    Horizontal wavelengths longer than ca. 25 km

Due to the FoV of GRIPS 9 at Kiruna, the instrument is sensitive to horizontal wavelengths of 25 km and longer. As mentioned in section 3.1, the OH*-equivalent (angular) BV frequency is calculated based on SABER temperature measurements in order to compute GWPED from GRIPS. Between day 1 and 60 of 2016, the OH*-equivalent (angular) BV frequency decreases overall. If one approximates the OH*-equivalent (angular) BV frequency linearly, the approximated

values range from ca. 0.022 1/s to 0.020 1/s (Fig. 5, solid line). However, superimposed fluctuations are visible, which reach ca. 13% deviation from the linear fit at maximum.

According to the data quality criteria for the GWPED calculation from GRIPS data (availability of at least 4 h of good quality data and exclusion of artefacts due to sunset and sunrise) as mentioned in Wüst et al. (2016), information about the energy content of gravity waves is derived between the nights from 13th to 14th and 30th to 31st January, 2016 based on GRIPS 9 measurements at Kiruna (as already mentioned above, the weather situation at ALOMAR did not allow the derivation of gravity wave information there).

For periods longer (shorter) than 60 min, the energy density varies between 10 and 160 J/kg (5 and 17 J/kg) with a mean of 43 J/kg (9 J/kg) (Fig. 6). Relative to these means, individual values lie within an interval of -77% and + 192% (-38% and +45%) for long (short) periods. The potential energy density of gravity waves with periods longer than 60 min can therefore be characterized as being more variable compared to periods shorter than 60 min. Fitting a cubic spline (non-iterative version as described in Wüst et al., 2017c) to the GWPED values suggests that a minimum of GWPED is observed around January 21st and 22nd and a maximum around January 27th for periods longer than 60 min. This overall behaviour cannot be confirmed for periods shorter than 60 min.

### 4.2.2     Horizontal wavelengths shorter than ca. 25 km

Wavelengths shorter than 25 km can be analysed using data of the airborne FAIM. The route of the first flight forms a triangle with the last two flight legs roughly parallel to a circle of longitude and latitude. The diagonal connection between Kiruna and approximately ALOMAR was covered three times in a row (Fig. 7). Although this flight track allows in principle the investigation of horizontal structures much larger than the FAIM FoV (via comparison of the individual flight legs), the airglow brightness varied too fast during the entire flight for achieving unambiguous results. This is especially apparent in the diagonal flight legs (Fig. 8a). Therefore, we concentrate on the analysis of horizontal structures in the range of the FoV size. As mentioned in section 2.2, the FoV is ca. 43 km x 55 km at 90 km height. However, its size changes with varying roll and pitch angles.

Figure 7 shows time difference images (time difference: 10 s) of the first flight. A difference image is derived by subtracting the intensity measured by each pixel from the intensity measured 10 s later by the same pixel. The velocity of the airplane is approximately 210 m/s. So, the airplane moves ca. 2 km in 10 s. Calculating a difference image emphasizes horizontal structures which change significantly within 2 km and/or within 10 s in flight direction. In the case of gravity waves, the result depends on the horizontal wavelengths and on the horizontal phase speed. For gravity waves with zero phase speed, constructive interference appears for a horizontal wavelength of 4 km in flight direction. The longer the wavelength, the less it will be emphasized. Destructive interference happens for horizontal wavelengths of 2 km (divided by integer factors) in flight direction. However, the above-mentioned constant shaking of the aircraft aids that also these small structures can at

least to some extent be identified in the images. So, one can say calculating a difference image is equivalent to applying some (high-pass) spectral filtering and amplification algorithm. The difference images of the first flight show wave-like structures of different wavelengths and amplitudes. There exists almost no region where such structures cannot be observed.

In order to derive quantitative results, the original (non-difference) images are analysed with a 2D FFT. Details about the different analysis steps are given in section 3.2. The shortest wavelength to which the FFT is sensitive is ca. 3 km (due to pre-processing with median blur applied to 17 pixels, 6 pixels correspond to 1 km). The FoV is reduced to 26 km × 26 km. It becomes clear that wave numbers and intensities vary in time and space (see Fig. 8b). In flight 1, high Fourier amplitudes also in the range of small-scale features (wavelength of 15 km wavelength and less) appear approximately between

16:20 UTC and 16:35 UTC (flight leg 1, turning manoeuver needs to be excluded) and between 17:30 and 18:00 UTC (flight leg 2 and 3). Between 18:30 UTC and 18:55 UTC (flight leg 4 and 5) especially small-scale features are relatively pronounced while larger-scale features are weaker compared to the time periods just mentioned. During these three time periods, the airplane was located east and/or southeast as well as above the Scandinavian mountain chain. The airglow brightness averaged over each picture shows local maxima during these three time periods (Fig. 8a).

Summarized over the whole flight, structures with wavelengths longer than 15 km propagate more frequently to the southeast (and/or northwest) than to the northeast (and/or southwest, Fig. 9). The majority of wavelengths shorter than 15 km moves to the northeast (and/or southwest). We identified 73 (31) events with wavelengths longer (shorter) than 15 km.

During the fifth flight on January 28[th], 2016, airglow observations were performed on the return from Karlsbad, Southern

Sweden, over Bergen, Southern Norway, along the coast line of Norway back to Kiruna (Fig. 10). The flight route can be divided into three legs. Several oscillations with 20–25 km are clearly visible at the beginning and at the end of the flight track. Especially, during the second (latitude-parallel) leg, many superimposed small structures with different orientations can be seen. The airglow intensity averaged over each image shows wave-like structures during each flight leg (Fig. 11a). They are most pronounced in flight leg 3.

Regions characterized by pronounced wave activity are observed especially after 18:30 UTC (Fig. 11b, mostly flight leg 3). During this time, the airplane flew along the coast line of Norway or above the Scandes. In contrast to flight 1, airglow brightness (averaged over each image) is maximal before the time period of maximal (Fourier) intensity. Flight 5 also differs from flight 1 concerning the propagation directions (Fig. 12): wavelengths longer than 15 km propagate mostly to the northeast (and/or southwest), for wavelengths shorter than 15 km a preferred quadrant of propagation cannot be

identified. A pronounced maximum can be found for eastward (and/or westward) propagation direction. In this direction nearly no larger-scale waves move. We identified 113 (63) events with wavelengths longer (shorter) than 15 km.

The occurrence rate of wave events varies during one flight and from flight to flight (Fig. 13). Overall, the legs of flight 5 show less variability than the legs of flight 1.

For both flights, the zonal legs (leg 5 of flight 1 and leg 2 of flight 5), where the airplane flew most of the time over the mountain chain and passed the highest elevations of the respective flight routes (grey line in Fig. 8a and 11a), are characterized by the highest occurrence rate (ca. factor 1.8–4.5 enhanced compared to the leg with the lowest occurrence rate of the respective flight). This agrees quite well with the visual inspection of the difference images of flight 5 (Fig. 10). For flight 1, this result is due to a large portion of small-scale wave-like structures (3–15 km wavelength) in leg 5.

The diagonal legs 1, 2 and 3 of flight 1 are identical concerning the route, however, the occurrence rate varies: it is highest in leg 2 and lowest in leg 3 (factor of ca. 2.6).

# 5    Discussion

Stratospheric winds varied strongly during January 2016 (Fig. 14). This was due to a minor stratospheric warming at the end of January (Dörnbrack et al., 2018). It was one of three consecutive minor stratospheric warmings which occurred before the

final breakdown of the polar vortex at the beginning of March 2016 (Manney and Lawrence, 2016). Starting mid-January 2016, the polar vortex became disturbed by planetary waves; especially planetary waves of zonal wave number 1 were amplified in the second half of January (Manney and Lawrence, 2016). Consequently, the vortex was displaced southward with its centre between Svalbard and Northern Scandinavia and the polar night jet became elongated in west-east direction (strong curvature over the Northern Atlantic and over Siberia, Dörnbrack et al., 2018). From January, 26[th] to February, 1[st]

2016, the meteorological regime above Kiruna was characterized by the transition of the stratospheric flow during the minor warming. After 30 January 2016, the horizontal wind in the stratosphere was rather light (< 20 m/s) and the stratopause was relatively warm (290 K, Dörnbrack et al., 2018). A stratospheric warming should affect the residual circulation and therefore the OH-layer characteristics. When stratospheric winds weaken or even reverse, filtering of gravity waves generated further down is changed. When there is a stratospheric wind weakening or reversal, the downward movement in the mesopause,

which is part of the residual circulation, weakens and this influences the OH excitation mechanism. However, planetary wave activity complicates this simplified picture: transmission of gravity waves is then a function of longitude (Whiteway and Carswell (1994), Dunkerton and Butchart (1984), and references in both publications).

We expect the following effect on the zonal means. The OH excitation mechanism is dominated by atomic oxygen which is produced at higher altitudes in the atmosphere (Shepherd et al., 2006). Processes which lead to vertical transport of atomic

oxygen influence the OH volume emission rate, but also height and thickness of the OH*-layer (see also, Liu and Shepherd, 2006; Mulligan et al., 2009; Grygalashvyly, 2015; von Savigny, 2015; Garcia-Comas et al., 2017). On average, height and thickness as well as height and intensity are anti-correlated. So, a weakening of the residual circulation should lead to a higher OH-airglow altitude, to a thinner OH-airglow layer, to a reduced OH-airglow intensity, and to a lower temperature. According to SABER measurements the temporal development of the airglow-layer characteristics observed at the end of

January and during February 2016 is consistent with the described expectations before and during a stratospheric warming.

Additional to the OH-layer characteristics, we also analysed periodic structures of different horizontal wavelengths. In literature, wave-like horizontal structures are often divided into ripples and bands. Ripple structures are phenomena with horizontal wavelengths of 5–15 km (Li et al., 2005, Taylor et al., 1995) or of 20 km at most (Takahashi et al., 1985). Their

lifetime is in the range of 45 min and less (Hecht, 2004). Fronts with large horizontal extent and wavelength, which can be sometimes observed for hours, are usually called bands (Taylor et al., 1995; Clairemidi et al., 1985). Hecht (2004) summarizes in his table 1 band and ripple characteristics based on four literature studies: the observed periods of ripples

(close to 5 min) are shorter than the ones for the bands, this also holds for the horizontal wavelengths. However, the provision of exact values does not seem to be possible.

In most studies, ripples are interpreted as signatures of local instability and may be related to or also be part of a breaking process of an atmospheric gravity wave (e.g., Li et al., 2005; Hecht, 2004; Fritts et al., 1997). In this case, they move with the background wind and can be separated into convective and dynamical ones according to their generation process (convective and dynamical instability, which occur for a Richardson number of less than 0 and 0–0.25, respectively). Ideally, the phase fronts of the dynamical (convective) ripples are oriented parallel (perpendicularly) to the associated gravity wave. However, also other cases have been observed (Hecht, 2004). Li et al. (2017) showed that more than half of the ripples they observed with an OH all-sky imager at Yucca Ridge Field Station, Colorado (40.7°N, 104.9°W), from September 2003 to December 2005 do not advect with the background winds and might not be instability features but wave structures that are hard to be distinguished from real instability features. In this case, the ripples could be related to the secondarily generated small-scale gravity waves (Vadas et al., 2003; Zhou et al., 2002).

In our airborne measurements, we find horizontal wavelengths in the range of ripples and bands. Concerning the latter we can argue that the chance to measure low-frequency (inertia) waves by FAIM is very much reduced compared to waves with higher frequencies. This can be deduced as follows. The higher the intrinsic frequency of a wave is, the smaller the angle between the wave fronts and the vertical must be. The horizontal wavelengths derived from the FAIM data are 26 km at maximum. Since the OH* layer extends over some kilometres, vertical wavelengths in the range of the full width at half maximum can barely be detected (Wüst et al., 2016). However, the vertical wavelengths of low-frequency waves are much smaller than the horizontal ones. Therefore, we can argue that FAIM is not very sensitive to low-frequency waves. GRIPS is less sensitive to short horizontal wavelengths than FAIM. In this study, the FoV of GRIPS is approximately 25 km x 25 km. Therefore, the signal of horizontal wavelengths in the range of 25 km and less is very much reduced or entirely averaged out. The argumentation concerning the resolvable vertical wavelength is the same as for FAIM. Therefore, GRIPS is less sensitive to high-frequency waves than FAIM.

Conclusions about possible sources of the observed bands are very difficult without a ray tracer since wind and temperature change with height influencing the angle of the wave front to the vertical. However, we can conclude the following.

Especially mountain waves (phase velocity equal to zero) generated near Kiruna had best chance to reach the OH-airglow layer around January 21[st] and from January 24[th] to 28[th]. During the other time periods, the horizontal wind speed became zero in extended height intervals (Fig. 14), which prohibits a vertical propagation (Fritts and Alexander, 2003). Indeed, we observed the highest occurrence rate of band-like structures in FAIM measurements during flight 5 (January 28[th]) when the airplane flew most of the time over the mountain chain and passed the highest elevations of the flight route.

The propagation of gravity waves with non-zero wind speed is discussed now. Before January 25[th], zonal winds in the upper stratosphere were stronger than in the upper troposphere above Kiruna (Fig. 14). Zonal winds in the stratosphere became

weaker after Jan 23$^{rd}$. During January 25$^{th}$ and 28$^{th}$, zonal winds in the troposphere and in the stratosphere were of comparable (eastward) velocity. So, for gravity waves generated in the troposphere, the stratospheric jet was not an additional filter and more waves should reach the OH airglow layer. After January 28$^{th}$, the vertical profile of the zonal wind showed regions of positive and negative wind velocity. An enhanced gravity wave filtering (for waves which had to pass the different regions) could therefore be expected. This agrees with the temporal development of the density of wave potential energy (periods longer than 60 min) derived from GRIPS, which depicts a maximum around the 27$^{th}$ of January (Fig. 6). It also agrees with the number of wave events observed by the airborne FAIM, which increased by a factor of 1.5–2.0 from flight 1 on January 14$^{th}$ to flight 5 on January 28$^{th}$ (the observation time of wave events, i.e. the time between the observation of the first and the last we event, changed only from 2.5 h to 2.9 h from flight 1 to flight 5). For gravity waves with periods shorter than 60 min, the temporal development of the GWPED derived from GRIPS measurements shows less and different variations. A significant maximum around January 26$^{th}$ might be present, at least compared to January 18$^{th}$/19$^{th}$ and 28$^{th}$.

A similar increase, in these cases of gravity wave activity or momentum flux, during a SSW is also reported by other authors. Yiğit and Medvedev (2012), for example, used a global circulation model in order to show that the activity of GW of lower atmospheric origin is enhanced by a factor of 3 in the course of the modelled warming. Based on WACCM (Whole Atmosphere Community Climate Model) Liu (2017) point out that the magnitude of the mesospheric momentum flux decreases significantly after the SSW event, but it varies strongly during the event. His findings agree with the observations of GW momentum flux changes during a SSW published by Wright et al. (2010), France et al. (2012), Thurairajah et al. (2014) and Ern et al. (2016), for example. An overview about the recent progress in understanding the role of gravity waves in vertical coupling during SSW is given by Yiğit and Medvedev (2016).

It is possible that the GW enhancement in the mesosphere is not only due to less filtering of tropospheric GW, the GW source could also be at higher altitudes. Liu (2017) argues that the rather rapid change of the winter jet system is expected to be a source of GW variability during SSW as GWs can be excited by imbalance of jet flow (O'Sullivan and Dunkerton, 1995; Zhang, 2004). Gerrard et al. (2011) find evidence that upward propagating gravity waves were generated in situ by a stratospheric temperature enhancement. However, as described above, the stratopause was relatively warm after January 30$^{th}$ (290 K, Dörnbrack et al., 2018) when our measurement period came to an end. The maximum in GWPED was observed earlier.

Compared to the zonal wind, the meridional component evolves differently (Fig. 14c): the direction of the meridional wind varies over the whole height range between January 20$^{th}$ and 27$^{th}$, 2019. Afterwards, this is not the case any more. If gravity wave filtering was driven by the meridional wind, one would expect that the activity of gravity waves generated in the troposphere increases at the end of January. This is not the case for our GRIPS observations.

Therefore, according to GRIPS data we conclude that the SSW affects gravity waves with periods longer and shorter than 60 min differently. The GWPED development for periods longer than 60 min is consistent with the assumption of a tropospheric source. However, we can also not exclude that at least parts of observed gravity waves are generated at higher

altitudes. The GWPED development for periods shorter than 60 min is less consistent with the assumption of a tropospheric source.

Based on the development of both horizontal wind directions it holds for the last third of January: the later in January, the smaller the difference between the frequency and the intrinsic frequency is. If the intrinsic frequency is approximately equal to the ground-based frequency, then the vertical wavelength does not vary much with height (only due to the changing (angular) BV frequency). If we assume that wave fronts of high- and medium frequency waves are oriented 0–45° to the vertical and if we take into account that our measurements address ca. 90 km height, then the possible tropospheric source must be ca. 90 km and more away from Kiruna.

In order to find out more about the different kinds of ripples, which we probably observed in the two flights, we need information about horizontal wind and temperature between 80 km and 100 km. Airborne or comprehensive ground-based measurements of these parameters are not available. Therefore, we do not get precise information about the background wind and about convective and dynamical instability along the flight track. We can therefore neither clearly distinguish between small-scale maybe secondary gravity waves and instability features nor (in the case of instability) between different kinds of instability.

From the temporal development of the (angular) BV frequency based on TIMED-SABER measurements (fig. 5), we can at least conclude that overall the tendency of the OH-airglow layer height to develop static instability increased during the measurement period. Since the measurements were taken in winter, the mesopause is located above the OH* layer at ca. 100 km height (e.g., Lübken and Von Zahn, 1991). Thus, as long as inversion layers do not exist, the vertical background temperature gradient is negative in the height range of the OH* layer. Static instability is therefore possible and independent of the existence of gravity waves.

In the following, we try to use the information about airglow brightness to learn more about convective and dynamic instability in the two flights. This argumentation only holds if the wavelengths in the range of ripples are instability features and if the airglow brightness variations (shown in Fig. 8a and 11a) are caused by gravity waves. We like to emphasize that we cannot prove these two conditions but our own (FAIM) data and also wind measurements at ALOMAR do not disagree with them. Concerning the airglow brightness variations, we can argue that the horizontal distance between the intensity maxima (compare Fig. 8a and 7 and Fig. 11a and 10) does not contradict the gravity wave possibility. Concerning the instability features, we have to check the background wind. As mentioned wind information for the flight is not available but mesopause wind measurements derived at ALOMAR are published (Stober et al., 2017). During flight 1 on January 14[th] ALOMAR was passed three times, flight 5 on January 28[th] took place south of ALOMAR. The zonal wind (after removal of tides and gravity wave contributions) is directed eastward on January 14[th] and 28[th] 2016 at ALOMAR. On January 14[th], the meridional component is nearly zero at ca. 86 km height, on January 28[th], it is positive (northward). Wavelength in the range of 15 km and less derived from airborne FAIM measurements move to the northeast (and/or southwest) during flight 1 (Fig. 9); for flight 5, a preferred quadrant of propagation cannot be identified, however, a maximum is observed for a strict

eastward (westward) propagation. So, we can at least say that the horizontal background wind at ALOMAR is not oriented perpendicular to the wave (ripple) propagation direction.

After having discussed the assumptions, we now come back to the actual argumentation. If gravity waves are present, then the probability for convective instability should change most in the grey regions of an airglow image (i.e., in the regions of average intensity) since the wave-induced absolute temperature gradient is maximal there. This can be explained as follows. The brightness of the OH* layer is mainly determined by the availability of atomic oxygen, which is generated higher up in the atmosphere (Shepherd et al., 2006). All processes which lead to a vertical transport of atomic oxygen therefore influence this parameter. Thus, downward transport processes are more pronounced in brighter areas of OH-airglow images compared to darker ones.

If the vertical transport processes happen adiabatically, e.g. due to a wave, adiabatic warming, i.e., positive temperature deviations from the atmospheric background, should be observed in the brighter parts of an airglow image. Darker airglow regions should then be affected by negative temperature deviations from the atmospheric background. Grey regions should be characterized by nearly no temperature deviation from the atmospheric background. We would like to emphasize here that a strict correlation between airglow brightness and temperature in the sense of the brighter the airglow, the higher the temperature and vice versa does not hold as for example Fig. 7 of Pautet et al. (2014) makes clear. However, at least during this time of the year this assumption holds on average (Garcia-Comas et al., 2017; Shepherd et al., 2006). So, if the vertical movements of atomic oxygen are due to a wave, one can conclude that the wave-induced vertical temperature gradient becomes zero where the brightness is maximal or minimal, while the grey regions can be interpreted as the zero-crossings of a wave observed in a vertical temperature profile. There, the steepest absolute vertical temperature gradient exists and the static stability of the atmosphere is most influenced.

The probability for dynamic instability should change most in the bright and dark regions of an airglow image. This follows from the gravity wave polarization equations: the zonal wind shear is maximized when the temperature is extreme (Heale et al., 2017).

As mentioned in the previous section, pronounced wave activity, also in the range of small-scale wave-like structures, is observed in bright airglow areas during flight 1 (ca. 16:20–16:35 UTC, 17:30–18:00 UTC, 18:30–18:55 UTC). If we assume, that these small-scale structures are ripples in the sense of an instability feature and that the generating gravity waves dominate the averaged airglow images, then this observation means that the ripples are mainly due to dynamical instability. For flight 5, the situation is different. Here, pronounced wave activity also of small-scale wave-like structures is not necessarily linked to very bright airglow areas. For example, at the beginning of our measurements, relatively low wave activity can be observed while the airglow brightness shows a broad maximum (Fig. 11). It is followed by a period (around 18:40 UTC–18:45 UTC) during which the airglow brightness is neither maximal nor minimal but relatively high wave activity is present. Therefore, our observations are consistent with the assumption that overall the importance of dynamical instability is smaller for this flight compared to the first one. However, as Li at al. (2017) point out the percentage of ripples which advect with background winds is ~30% in both summer and winter in their data basis, which is probably the largest

one investigated with respect to this phenomenon. This number is much lower than expected. Therefore, the probability is high that also in our case a large part of the observed ripples are not instability features.

# 6    Summary and conclusion

Wave-like structures in the range of ripples and bands, ca. 3–25 km horizontal wavelength, were observed during two selected flights (flight 1 on January, 14[th], 2016 and flight 5 on January, 28[th], 2016) of the airborne airglow camera FAIM in Northern Scandinavia. The flights were separated by 14 days and took place under different atmospheric conditions: while the stratospheric jet was rather strong during the first flight, it was much weaker during the last one. In the same time, ground-based airglow observations (temperature and intensity) as well as TIMED-SABER based measurements (OH-airglow layer height and thickness) revealed typical features of a stratospheric warming.

The activity of these wave-like structures depended on place and time. Regions of vanishing wave activity were not observed. The most mountainous regions were characterized by the highest occurrence rate of wave-like structures in both flights. For flight 1, this result is due to a large portion of structures in the range of ripples. At the time of this flight, the propagation of mountain waves was not possible or at least strongly reduced. This is probably not the case for flight 5.

The static stability of the airglow-layer height decreased during January based on TIMED-SABER. If one interprets ripples as instability features, the investigation of the airborne FAIM data shows consistent results .

The wave potential energy referring to waves of ca. 25 km horizontal wavelength and more varied also in time (time period: 14[th]–30[th] January) as ground-based airglow observations by GRIPS combined with SABER data during January 2016 showed. For waves with periods longer than 60 min, it is characterized by signatures which would be expected for waves generated in the troposphere. Periods shorter than 60 min evolve differently. Therefore, we conclude that wave events with periods longer than 60 min are generated at different heights than waves with periods shorter than 60 min.

**Author contribution**

Carsten Schmidt, Patrick Hannawald and Sabine Wüst assured the operability of the ground-based and airborne instruments, GRIPS and FAIM, during the GW-LCYCLE campaign. Martin G. Mlynczak and James M. Russell are responsible for the TIMED-SABER data. Patrick Hannawald, Sabine Wüst and Carsten Schmidt analyzed the data. Sabine Wüst and Michael

5 Bittner formulated the respective research goals in the proposal GW-LCYCLE. Sabine Wüst wrote the manuscript and discussed it especially with Michael Bittner, Patrick Hannawald and Carsten Schmidt.

**Competing Interests**

The authors declare that they have no conflict of interest.

**Acknowledgement**

For the overall organization of the flight campaign, we thank our colleagues from DLR, especially Markus Rapp and Andreas Dörnbrack, institute of atmospheric physics (IPA). For the acquisition of the project GW-LCYCLE, we thank Markus Rapp. Verena Wendt and Jeng-Hwa Yee deserve gratitude for her preparatory work concerning the SABER data

15 analysis.

We thank the German Ministry for Education and Research (BMBF, Grant agreement No: 01LG1206A) for funding. Some algorithms applied in this study were developed in the VAO-project LUDWIG which was funded by the Bavarian State Ministry for the Environment and Consumer Protection (project number TUS01 UFS-67093).

Processing and long-term archiving of the FAIM and GRIPS data is provided by the World Data Center for Remote Sensing

20 of the Atmosphere (WDC-RSAT, http://wdc.dlr.de). The measurements are part of the Network for the Detection of Mesospheric Change, NDMC (https://www.wdc.dlr.de/ndmc).

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

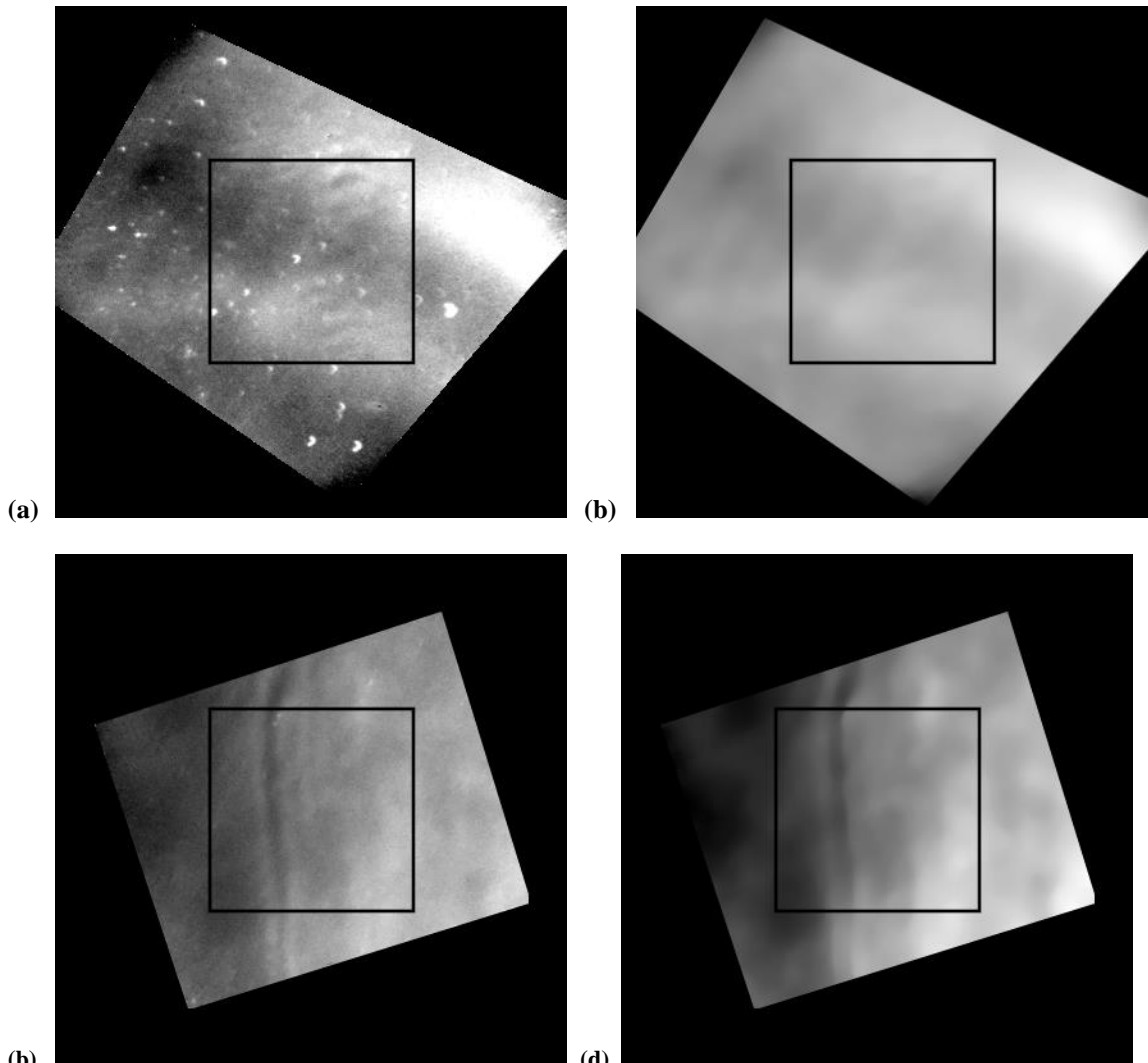

**(a)**

**(b)**

5 **(b)**

**(d)**

**Figure 1: Images from the airborne FAIM 2 during the first flight. The left column (a and c) shows two images which are flat fielded, contrast adjusted, un-warped (due to pitch, roll and yaw angle), rotated to a northward position and mirrored to fit the correct west-east position giving a satellite's view of the airglow layer. The right column (b and d) shows the same images after 10 applying the median blur. The image in the upper row (a and b) is an extreme example since the roll angle of -27° is rather high which results in a large image. For the analysis only absolute roll angles of 25° at maximum are used. This holds for the image shown in the second row (c and d). The squares mark the regions of interest which are used for the calculation of the spectral analysis if roll and pitch angles fulfil the selection criteria.**

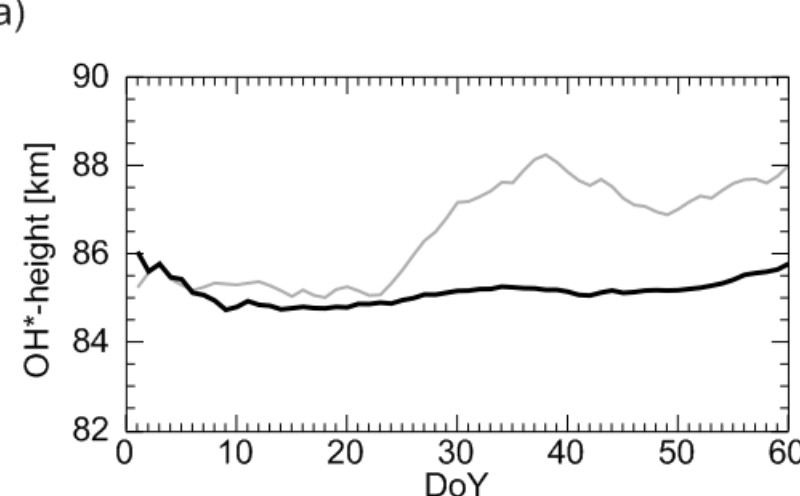

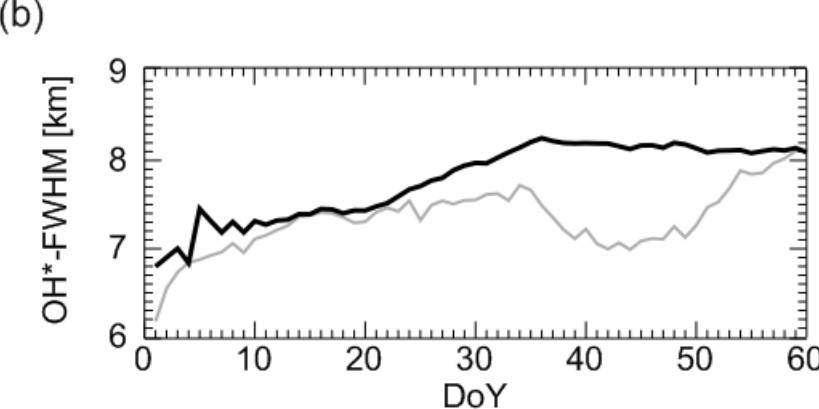

**Figure 2: The different curves represent 15-days running means of the OH\*-layer height (a) and its FWHM (b) calculated from TIMED-SABER VER profiles (OH-B channel) within a rectangle centred at ALOMAR with 300 km edge length. Shown is the time period DoY 1-60. The black line is the mean of all years, the grey line the year 2016. If more than one value is available per day, a daily mean is calculated before smoothing. However, it can also happen that no SABER measurement is available for a specific day.**

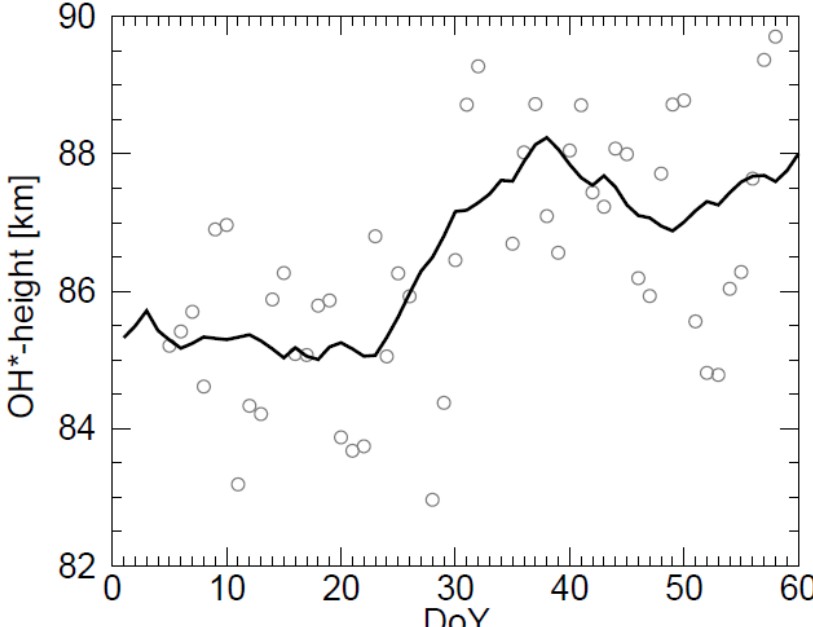

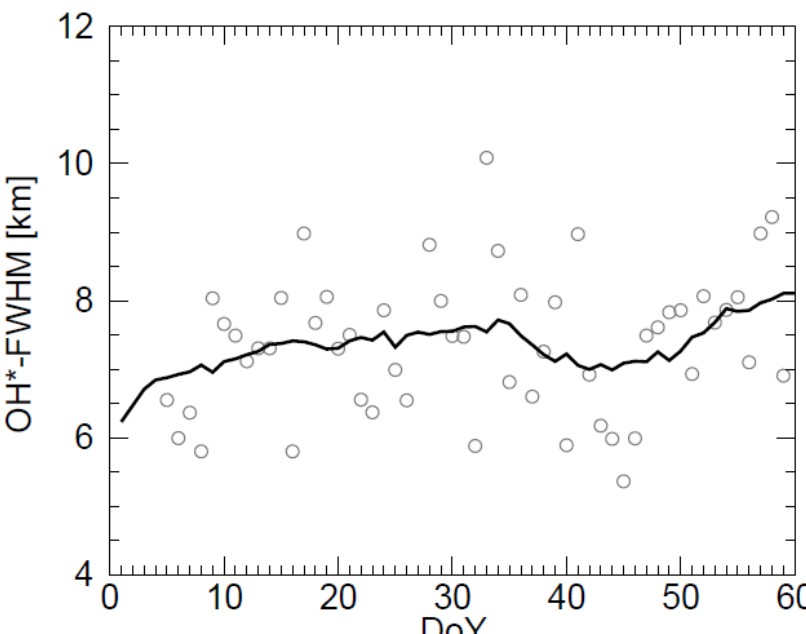

Figure 3: As in Figure 2, the solid lines represent 15-days running means of the OH*-layer height (a) and its FWHM (b). The circles stand for the individual values. Shown is the year 2016.

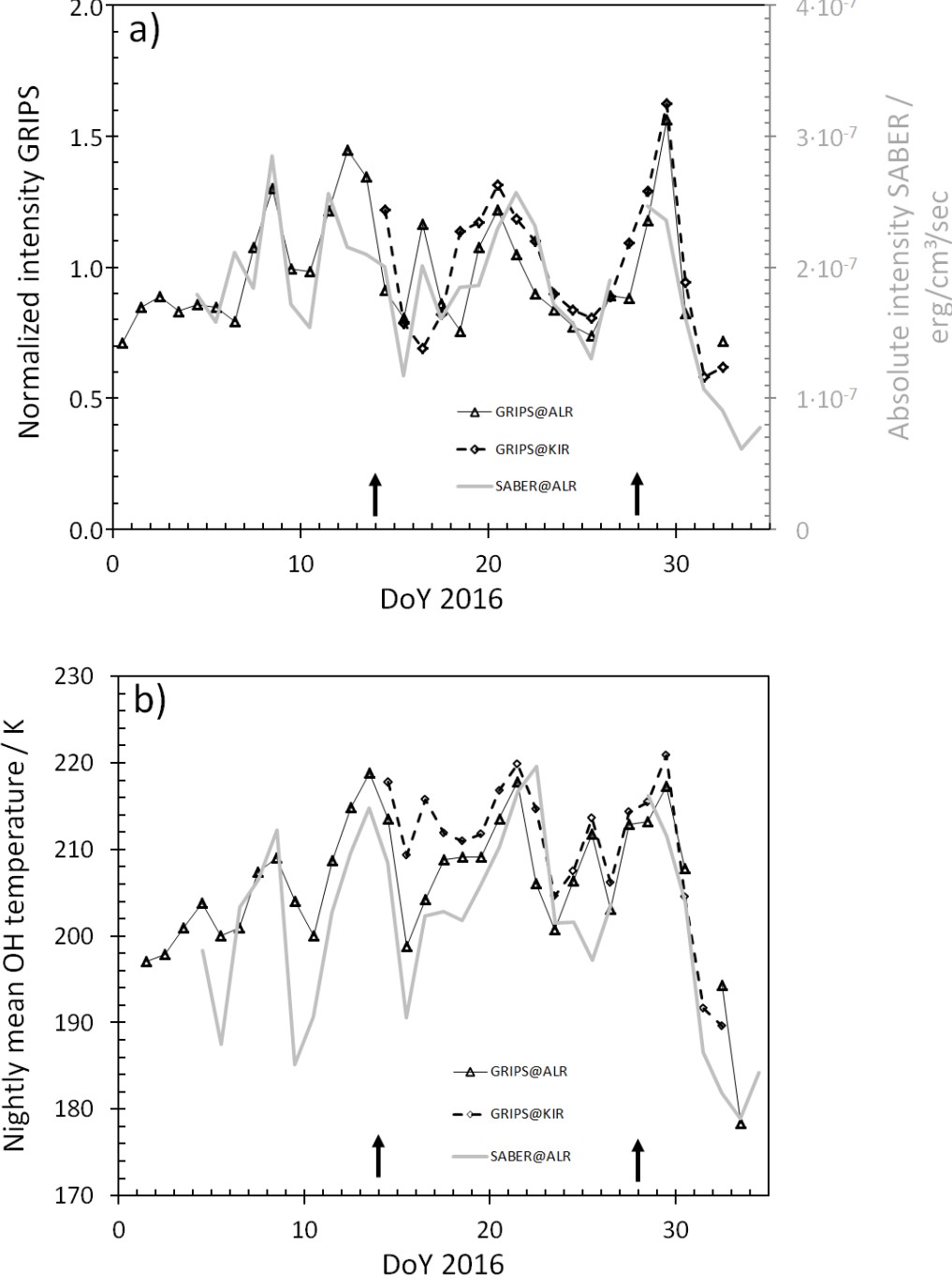

**Figure 4: a)** Average airglow intensity during the GW-LCYCLE campaign, covered by the three instruments GRIPS 14 at ALOMAR (solid, triangles), GRIPS 9 at Kiruna (dashed, diamonds) and SABER (grey) normalized to the respective mean intensity of the time period (1st of January to 3rd February 2016). The arrows mark the dates of flight 1 and 5. **b)** Nightly mean OH temperatures at ALOMAR (solid) and Kiruna (dashed). The temperature values agree within ca. 6 K on average. This is one order of magnitude lower than expected for this latitude, distance, and season according to figure 5 from Wendt et al. (2013).

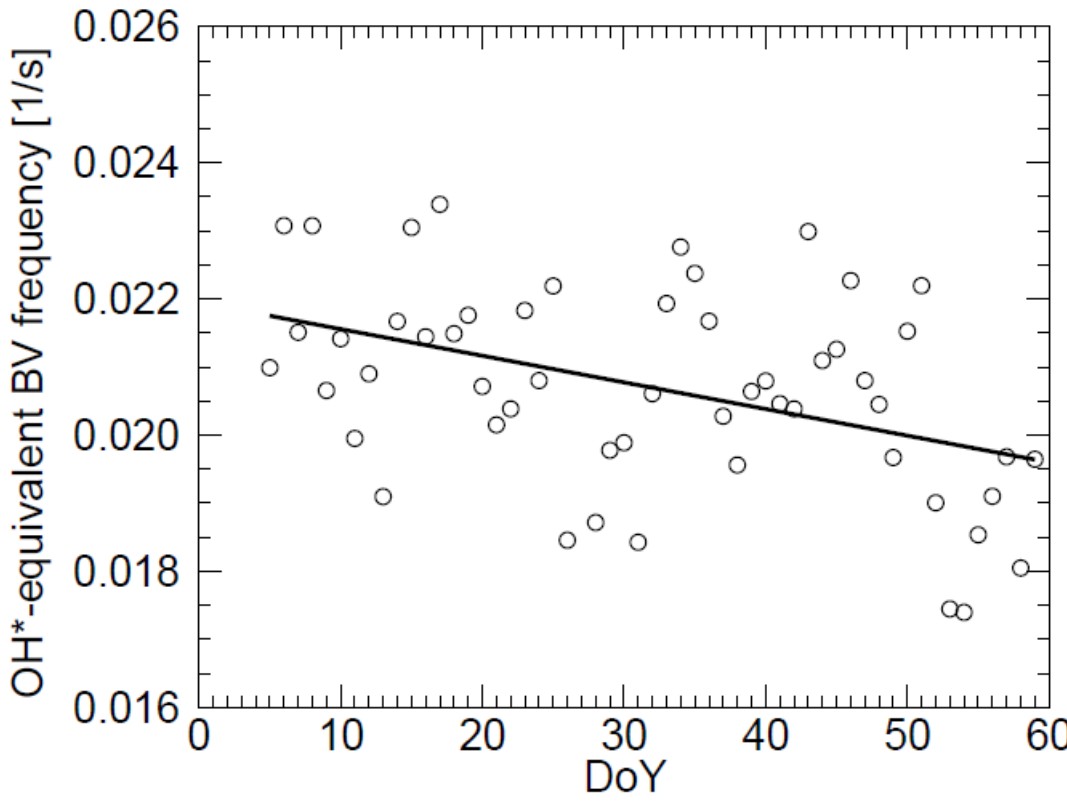

**Figure 5: Overall, the OH\*-equivalent (angular) Brunt-Väisälä (BV) frequency (circles are individual values) derived from SABER decreases from DoY 1–60 2016 over ALOMAR. The mean difference between the linear fit and the daily OH\*-equivalent (angular) Brunt-Väisälä (BV) frequency is ca. 5%.**

**(a)**

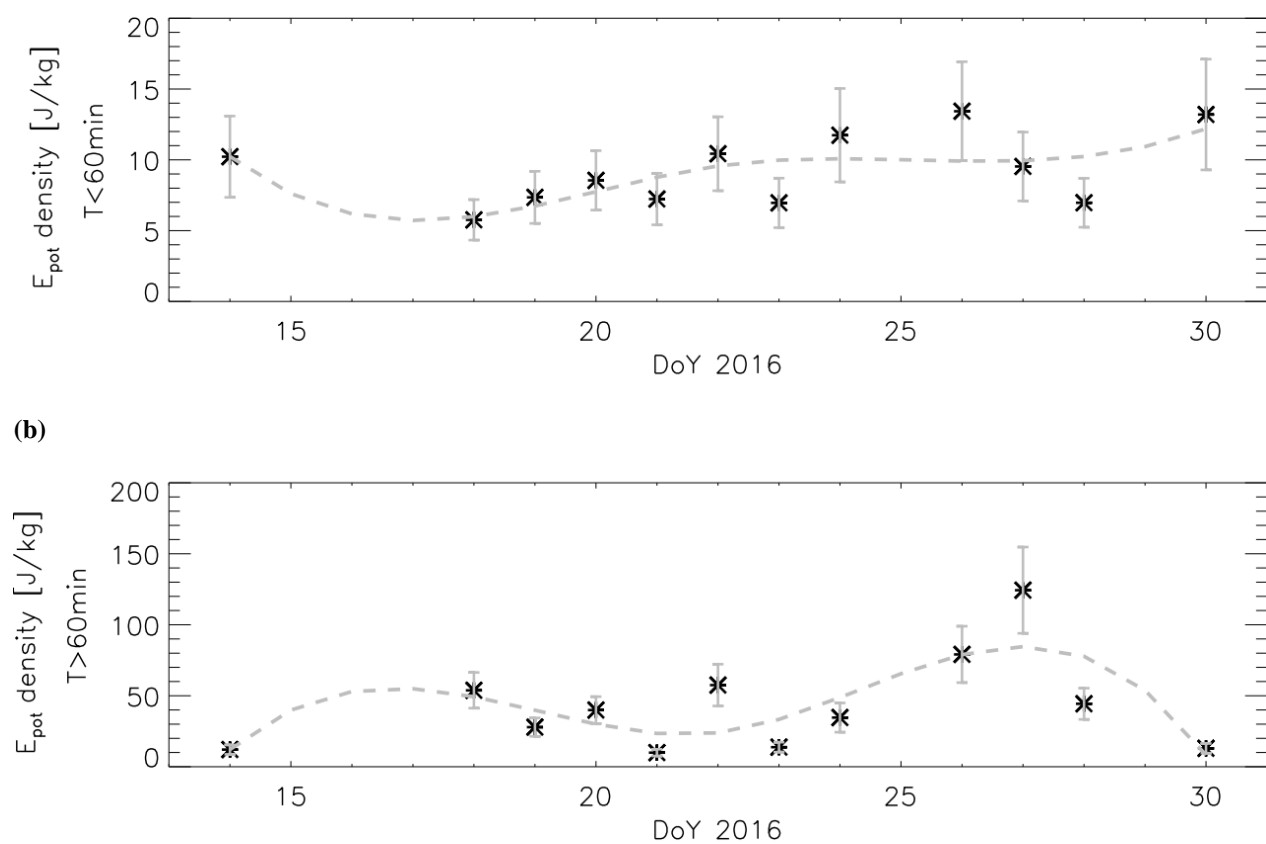

**(b)**

Figure 6 The upper plot shows the nightly mean wave potential energy density (based on GRIPS 9 measurements at Kiruna) for periods shorter than 60 min, while part b refers to periods longer than 60 min. The nightly mean OH*-equivalent (angular) BV frequency was taken from SABER. For the night from January 27[th] to 28[th], coincident SABER profiles were not available, therefore, the mean based on the values of the night before and after was calculated. A cubic spline approximation is superimposed (dashed line).

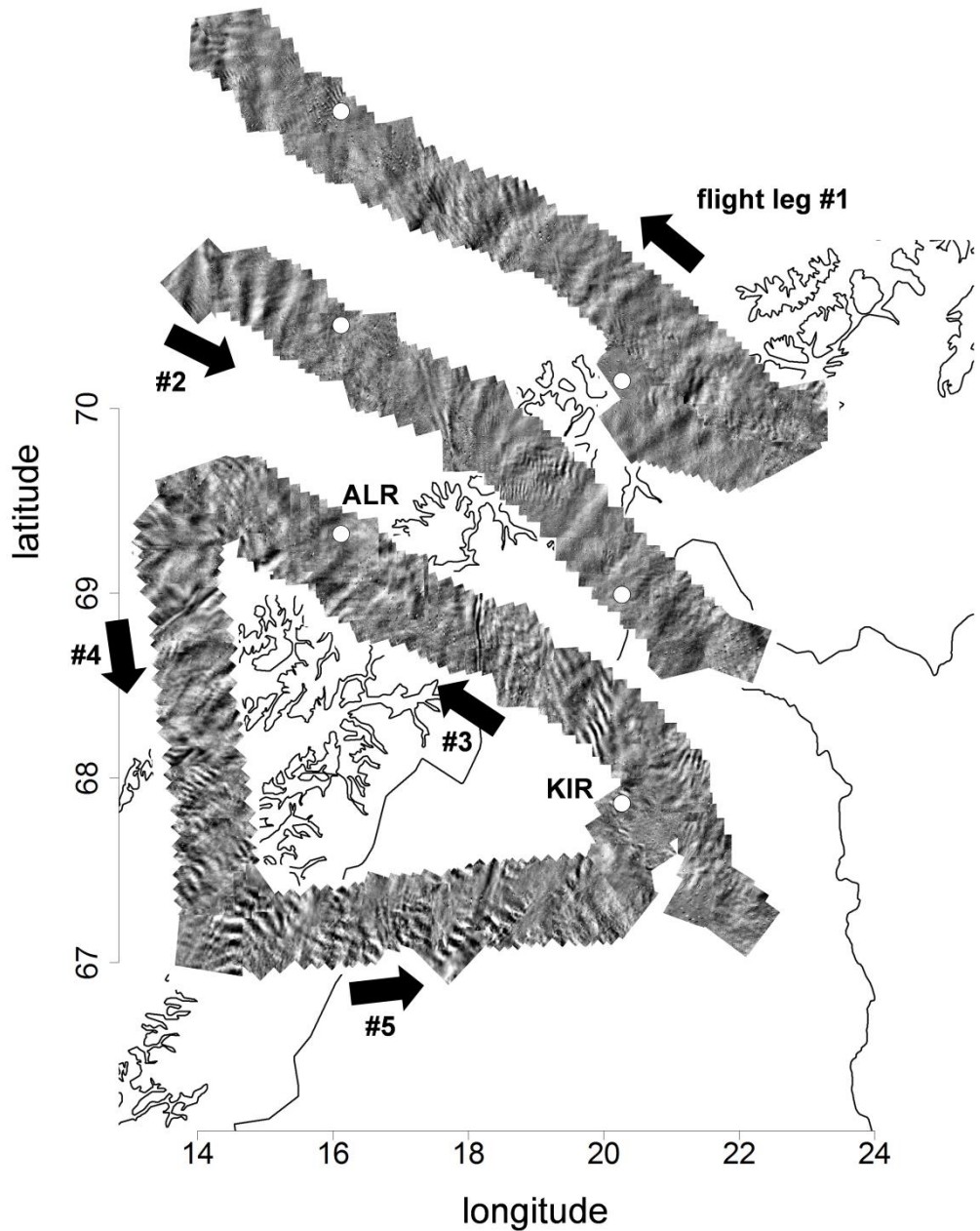

**Figure 7: Difference images of the first flight (Jan 14). A difference image is derived by subtracting the intensity measured by each pixel from the intensity measured 10 s later by the same pixel. The velocity of the airplane is approximately 210 m/s. So, the airplane moves ca. 2 km in 10 s. Calculating a difference image emphasizes horizontal structures which change significantly within 2 km and/or within 10 s in flight direction. Please note that the first three legs cover the same area, but legs one and two have been shifted for a better display. The black arrows show the flight direction. Apparently, the small scale structures change rapidly within a few minutes.**

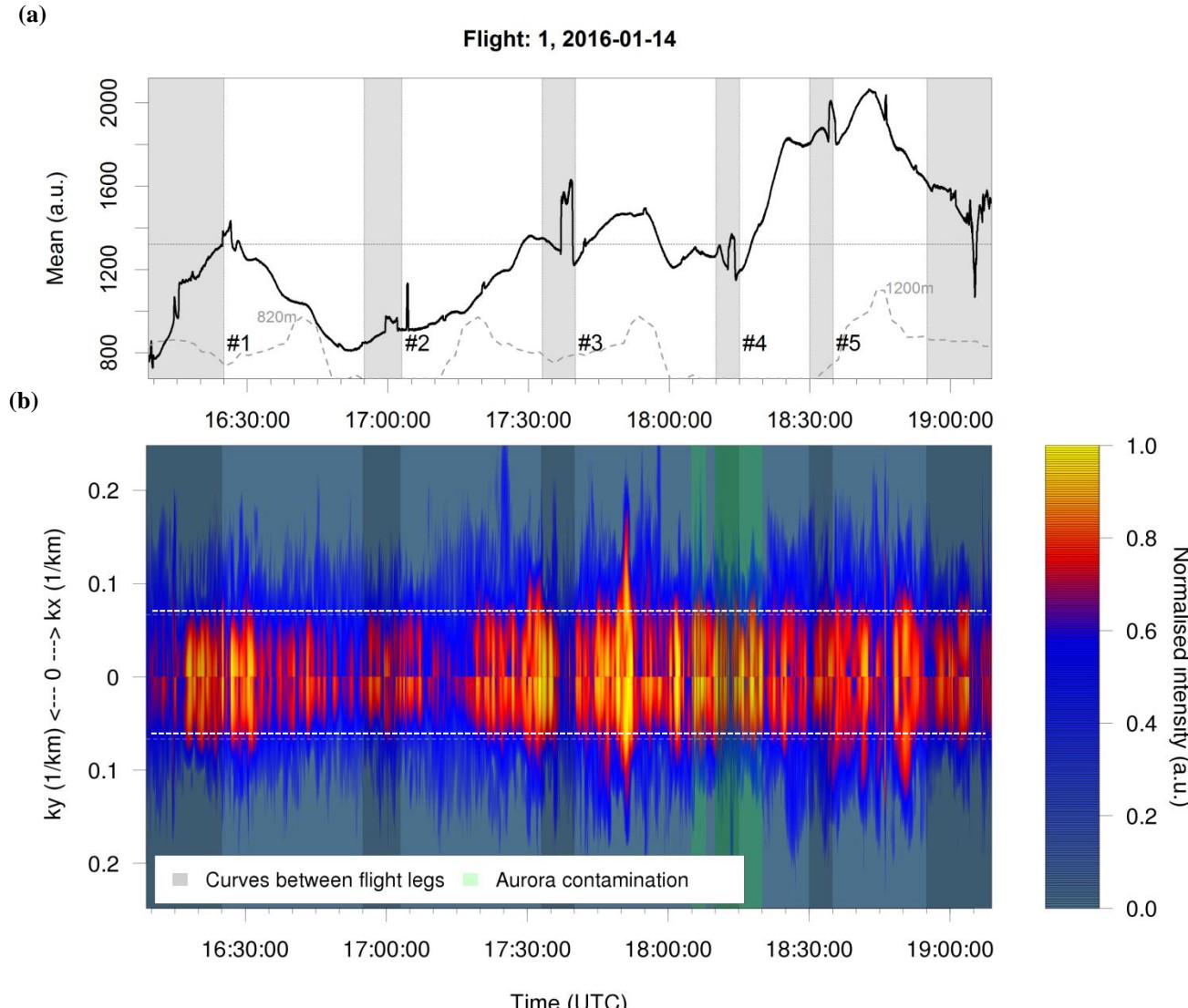

**Figure 8 (a)** Time series of integrated intensity per FAIM image. The grey areas refer to turning manoeuvres and should be excluded from further analysis. The grey line shows the orography. **(b)** 2D FFT spectra versus time: in the upper (lower) part the spectral intensity depends on the zonal (meridional) wave number. This plot is created by summing up the significant spectral intensities over the meridional (zonal) wave numbers for each image. The colour bar is normalized in a way that the different spectra are comparable within one flight (logarithm to the basis of 10 is applied to each spectrum, mean and standard deviation over these values of the whole time series are calculated, values higher or lower than the mean plus or minus two times the standard deviation are set to 1 or 0). The horizontal line marks the wavelength of 15 km.

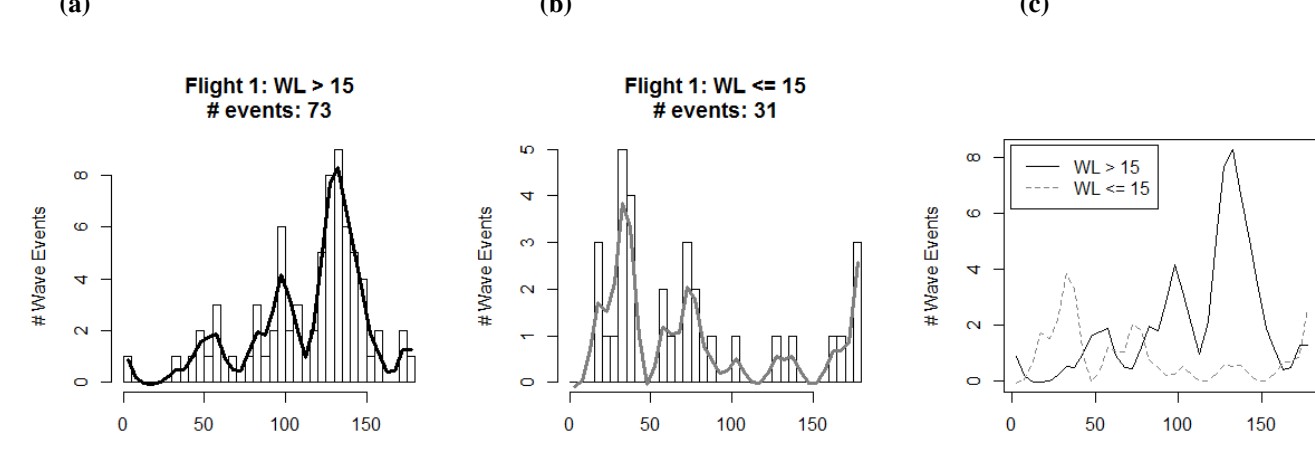

**Figure 9** If wavelength and propagation direction are identical for at least six images separated by 30 s at maximum (parameter n), this wavelength is denoted as wave event. Wave events must be present for more than 10 s at least (time difference between first and last occurrence, parameter t). a) and b) depict the histograms of propagation directions (180° ambiguity, 5° bar width) smoothed by a cubic spline for wave events with horizontal wavelengths longer and shorter than 15 km for flight 1. Part c) shows both splines in one plot (black and grey: wavelength longer and shorter than 15 km). Smaller n and t change the absolute values for wavelengths longer than 15 km but not the qualitative results. Wavelengths shorter than 15 km are more sensitive to these parameters (especially to n) but with a stable peak at ca. 40°.

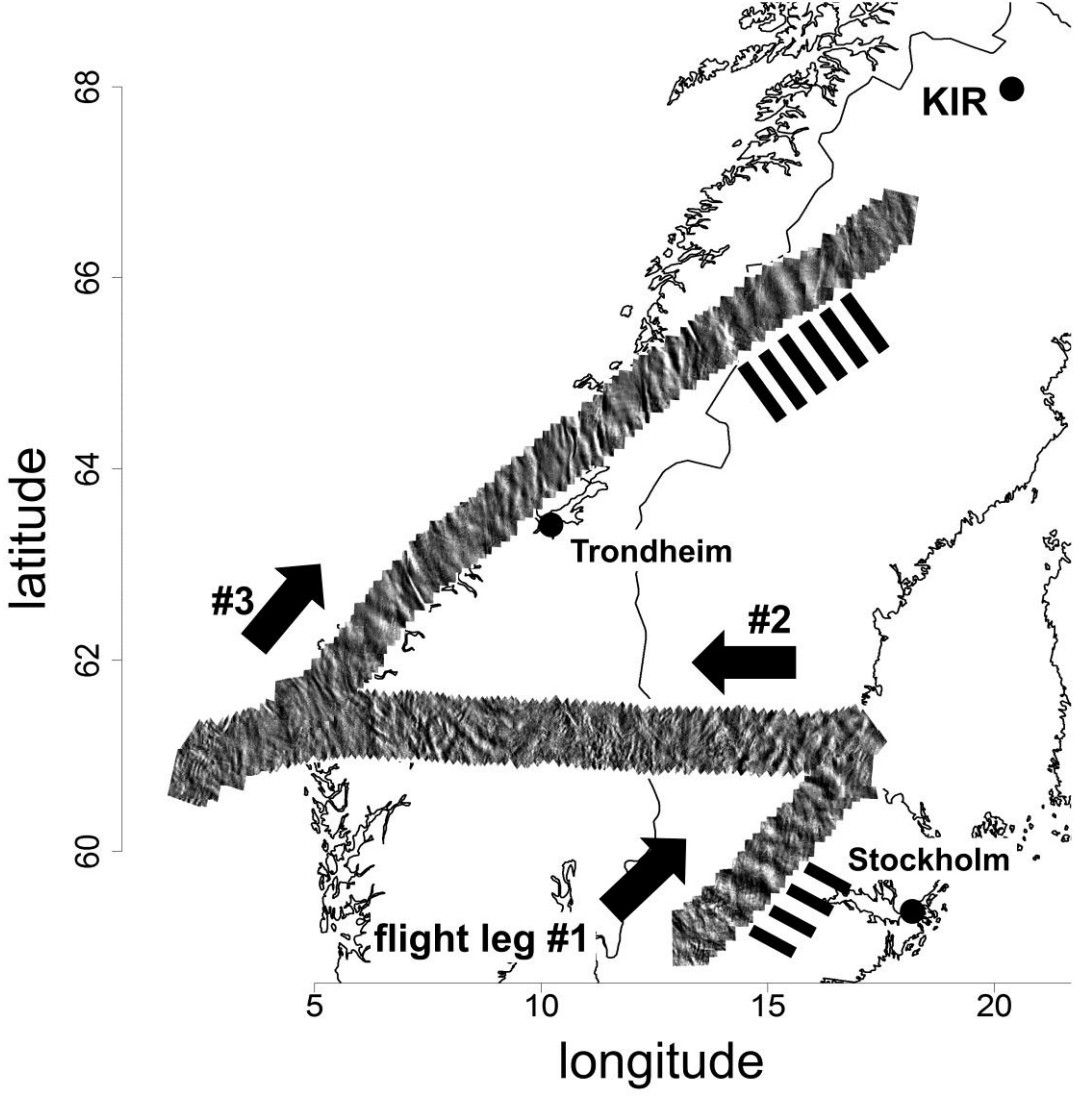

**Figure 10: Same as Figure 7 for the second flight of Jan 28, starting in Karlsbad. During the descent to Kiruna, high clouds obstructed the FoV and the respective observations are not shown. The black bars denote the appearance of gravity waves with larger scales than the instantaneous FoV.**

**(a)**

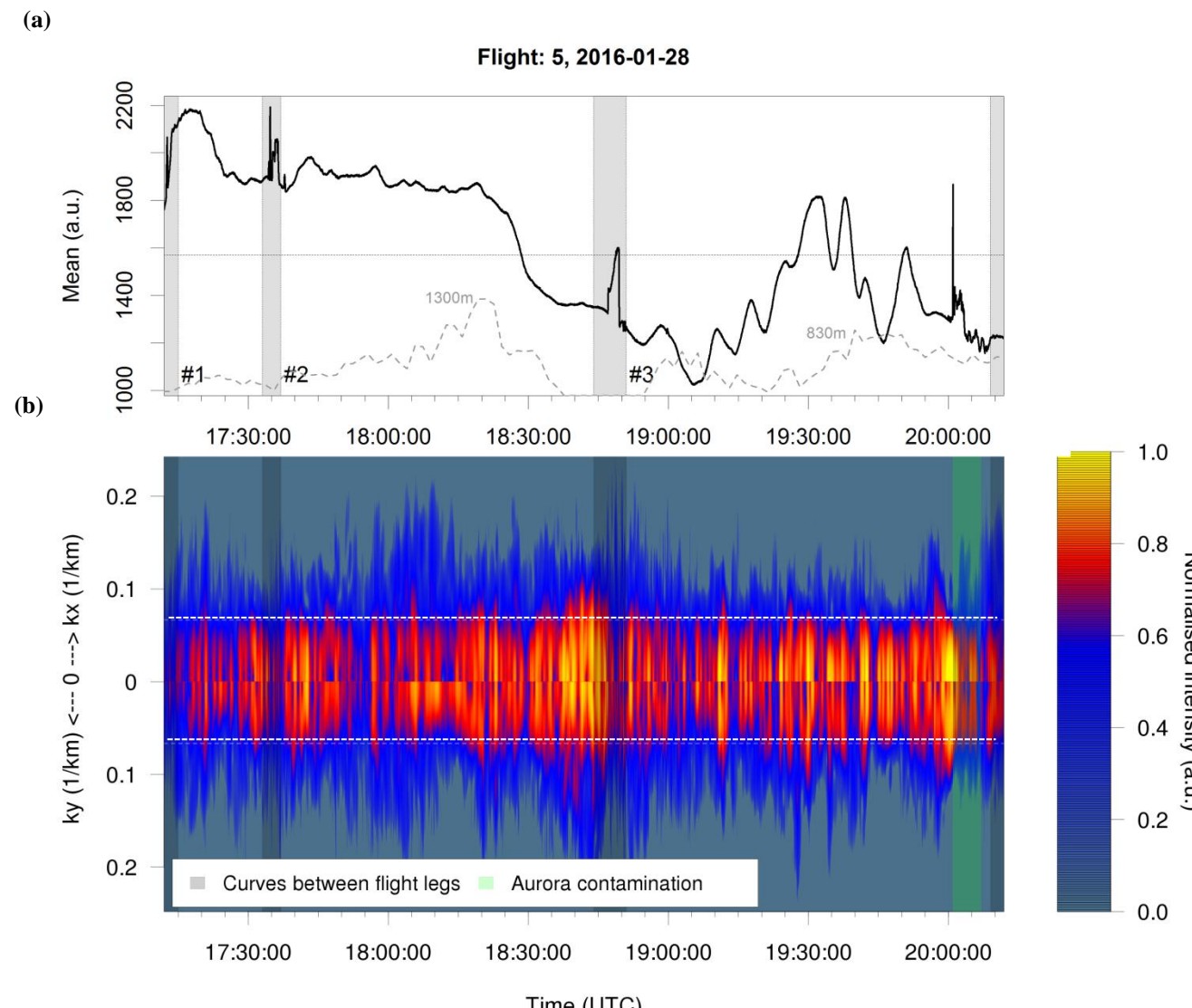

**(b)**

Figure 11 Same as Figure 8 but for flight 5.

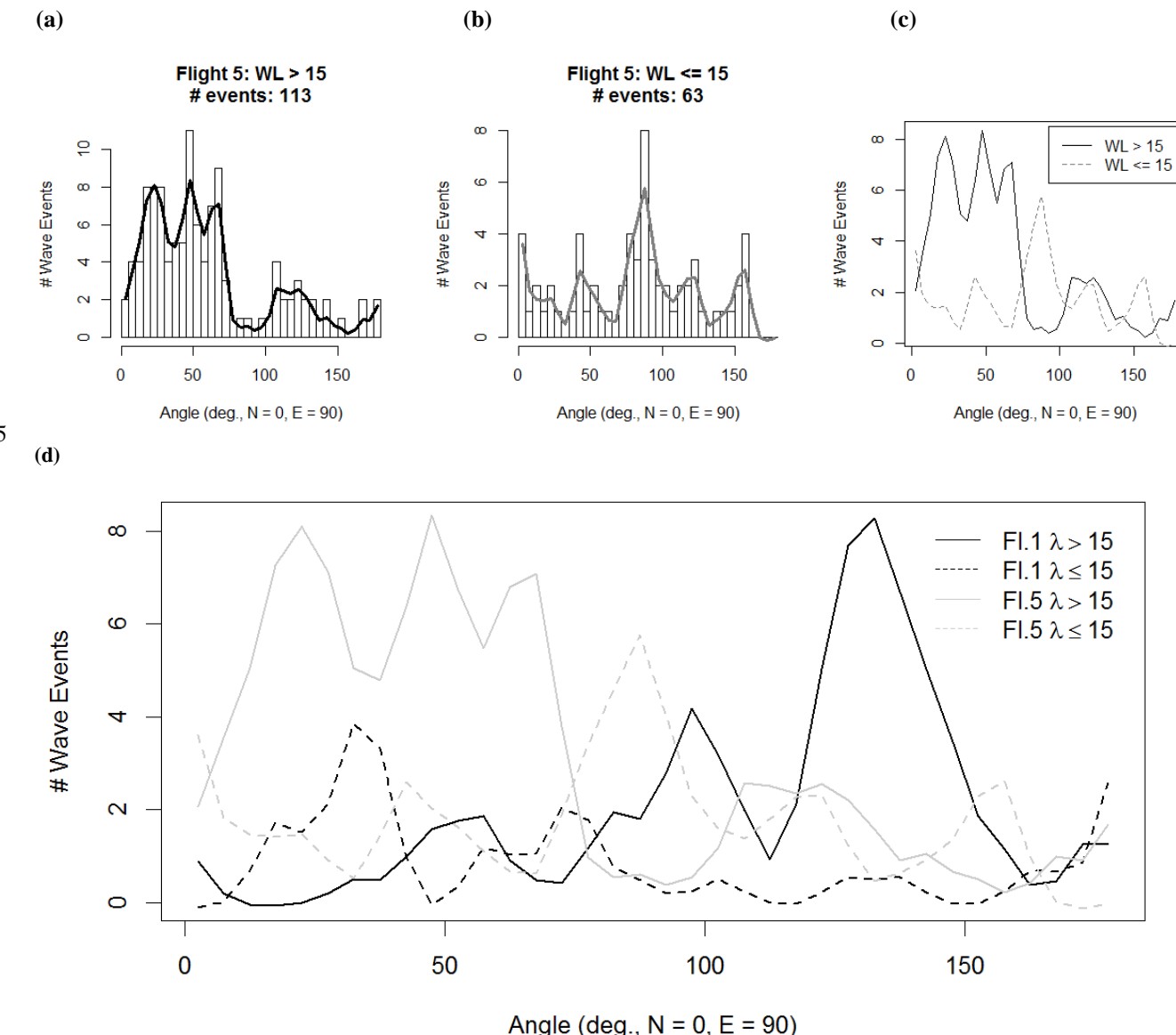

**Figure 12 same as Figure 9 but for flight 5. In this case, the results for both wavelength ranges are qualitatively stable for different n and t (the smaller n and t, the higher the absolute values). Part (d) shows the direct comparison of the results of both flights.**

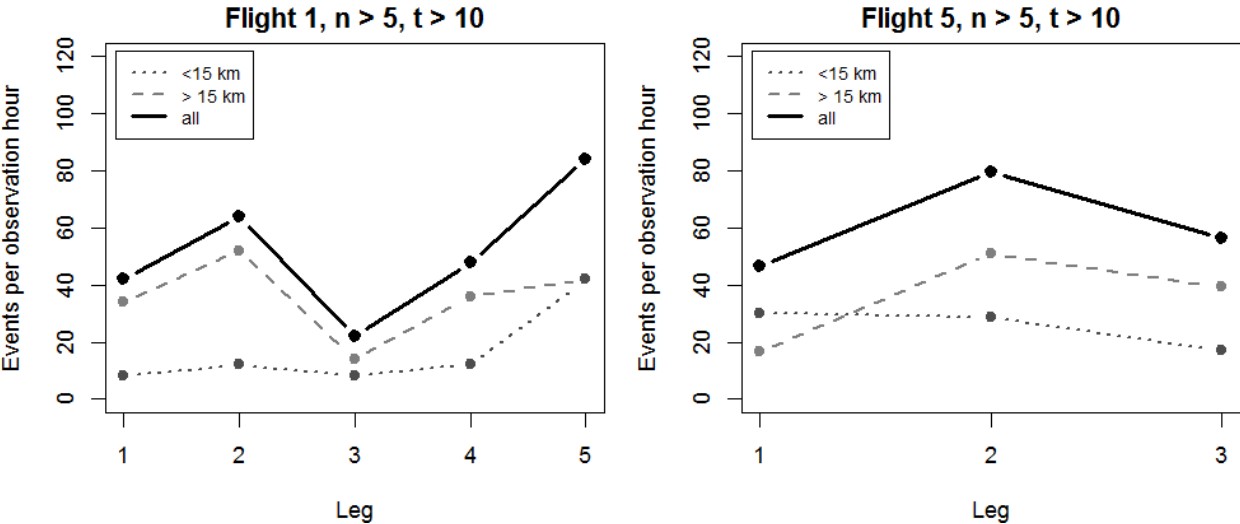

**Figure 13** Occurrence rate of significant wave events for each leg of flight 1 and 5 (measurements during turning manoeuvres and aurora events are not included). The results are shown for wavelengths smaller (dotted line) and larger than 15 km (dashed line) as well as for the sum of both (solid line). If wavelength and propagation direction are identical for at least six images (parameter n) separated by 30 s at maximum, this wavelength is denoted as wave event. Wave events must be present for more than 10 s at least (time difference between first and last occurrence, parameter t). The results do not change qualitatively if n and t are smaller.

**(a)**

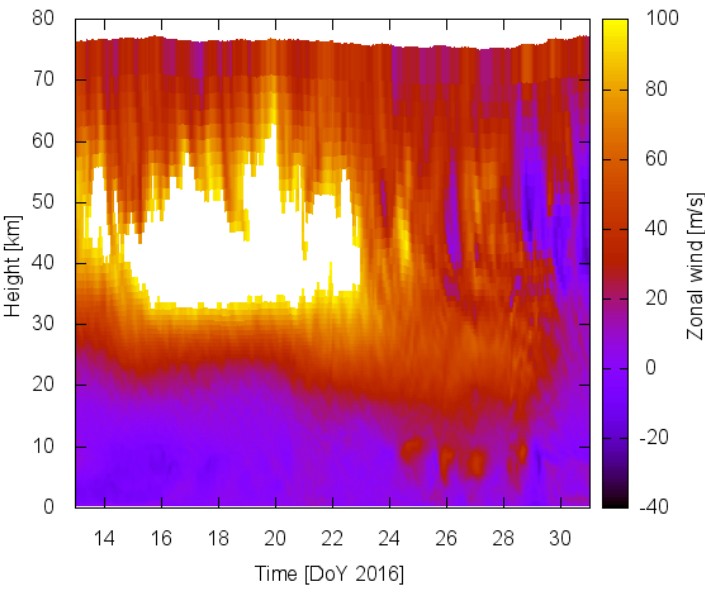

**(b)**

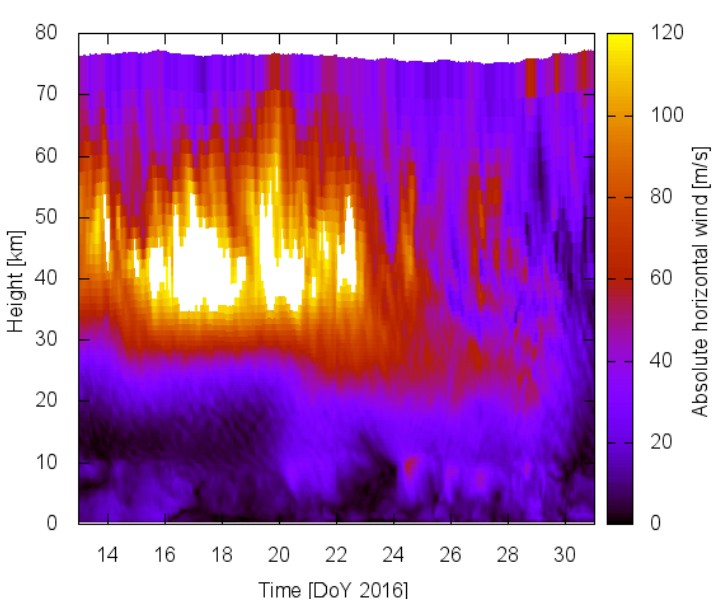

**(c)**

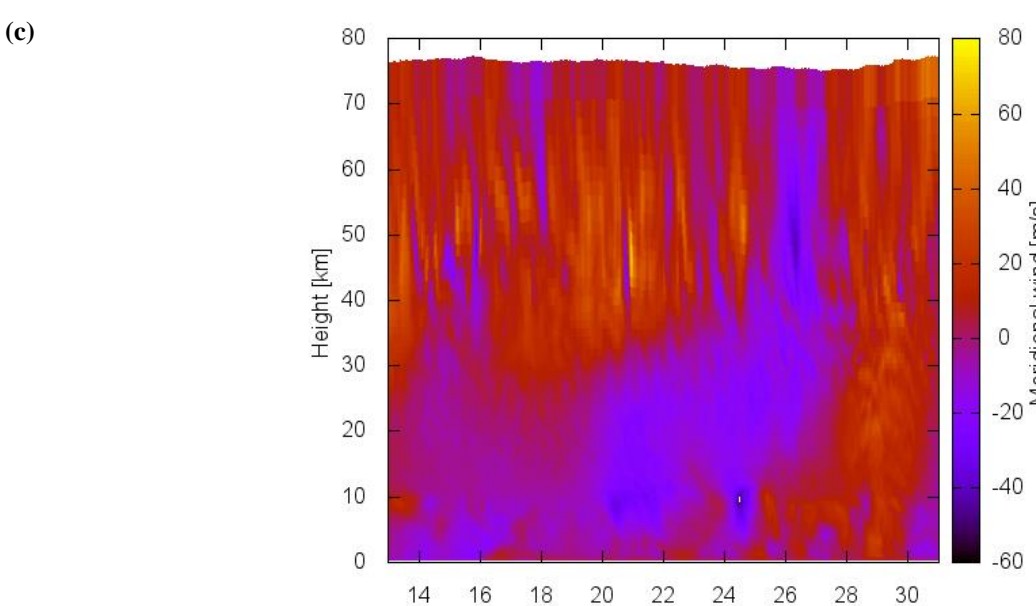

**Figure 14 Zonal (a), absolute horizontal (b), and meridional (c) wind at 67.84° N, 20.41° E (Kiruna: 67.86° N, 20.24° E) from ECMWF data (European Centre for Medium-Range Weather Forecast, ECMWF data were provided by Andreas Dörnbrack, DLR).**