# Peer review of "Observations of OH-airglow from ground, aircraft, and satellite: investigation of wave-like structures before a minor stratospheric warming"

_Atmospheric Chemistry and Physics, 2018_

## Referee Comment (RC1) · Anonymous Referee #1 · 29 Dec 2018

General comments:

This manuscript describes and analyzes observations of OH airglow emissions from different platforms (particularly from the FALCON aircraft) during a field campaign in Scandinavia in January 2016. The paper is in general well written and easy to follow. Although it does not provide any really new insights into the topic, the manuscript should in my opinion by published. It will probably be complemented by other publications dealing with the same field campaign. I recommend accepting the manuscript subject to minor revisions. Below I offer some (mainly really minor) suggestions for

[Figure]

improvements.

Specific comments:

Page 1, line 29: 'emphasize' -> 'emphasis'

Page 2, line 12: 'mounted at' -> 'mounted on' ?

Page 2, line 14: 'all other airborne measurements address heights of ca. 20 km and below.'

It's not entirely clear, what this part of the sentence refers to? To other instrumentation on Falcon?

Page 3, line 25: I suggest replacing 'x' in '320 px x 256 px' by '\times' (I assume you use LaTex? )

Page 3, line 30: '(compare Fig. 7 and 10).'

not sure, how this can be seen in Figures 7 and 10? Do the arrows indicate the flight track? The observed area is sometimes left, sometimes right of the arrows.

Page 4, line 22: 'and upper levels'

It's not entirely clear, what 'upper levels' refers to. It may refer to 'upper atmospheric levels'or 'levels of excitation'. It's most likely the latter. Please specify.

Page 4, line 24: delete comma before '1.04 and 1.06'

Page 4, lines 28 – 30: Perhaps a paper on the validation of SABER temperature observations can be cited here?

Page 6, line 19: 'information are' -> 'information is'

Page 6, line 23: 'the height of 84 km'

Is this the height of maximum VER or a weighted, i.e. centroid altitude?

Figure 3: It would be good to mention explicitly in the Caption of Figure 3 that the year 2016 is shown.

Page 8, lines 24/25: 'information .. are' -> 'information .. is'

Page 9, line 7: 'flight legs parallel'

Perhaps add, e.g. 'roughly' or 'more or less', because the flight legs do not appear to be exactly parallel to the latitude/longitude circle?

Caption Figure 7: What exactly do you mean with 'Difference images'? This is later explained in the main text - as I found out - but perhaps it can be explained briefly in the caption, too?

Page 11, line 15: 'as well as height and intensity are anticorrelated.'

Regarding the anti-correlation between intensity and emission altitude the papers by Grygalashvyly (2014) and von Savigny (2015) may be cited, too. The first one provides a theoretical explanation for this anticorrelation and the second one shows the relationship for the OH(3-1) band (if I remember correctly), which is of importance to your work.

Grygalashvyly, M., G. R. Sonnemann, F.-J. Lübken, P. Hartogh, and U. Berger (2014), Hydroxyl layer: Mean state and trends at midlatitudes, J. Geophys. Res. Atmos., 119, 12,391–12,419, doi:10.1002/ 2014JD022094.

von Savigny, C., Variability of OH(3-1) emission altitude from 2003 to 2011: Long-term stability and universality of the emission rate - altitude relationship, J. Atmos. Sol.-Terr. Physics, 127, 2015.

Page 13, line 19: 'So, if the vertical movements of atomic oxygen are due to a wave, one can conclude that the wave-induced vertical temperature gradient becomes zero where the brightness is maximal or minimal,'

I don't really understand this argument. Please explain.

[Figure]

Caption Figure 8: 'The horizontal line marks the wavelength of 15 km.'

There are different horizontal lines. Please specify.

---

## Referee Comment (RC2) · Anonymous Referee #2 · 3 Jan 2019

General Comments The authors present a selection of results from a campaign of observations on OH-airglow emissions recorded from the ground, and from an aircraft flown inside the Arctic Circle during January and February 2016. The ground-based observations were made using infrared spectrometers deployed at ALOMAR and Kiruna, while the aircraft measurements were made with a "Fast" airglow imager taken over flight paths that included both observing stations.

These observations were supplemented with the inclusion of temperature and OH* VER profiles from the TIMED-SABER satellite instrument near the time and location of

the observations, in addition to horizontal and meridional wind data from the ECMWF.

The SABER data have been used to calculate the mean height and thickness of the OH* layer during the period of the observing campaign. There is good correspondence between the variation of the OH* layer brightness measured by the ground based spectrometers and that obtained from the SABER VER measurements. The Brunt-Väisälä (BV) frequency during the observing campaign was calculated for the OH* layer by weighting it by the VER measurements. It showed a steady decrease throughout the observing period – which was interpreted to imply a reduction in the static stability of the atmosphere during that tim interval.

Combining the OH*-layer averaged temperature data from the infrared spectrometers with the SABER temperature profiles, enabled the authors to calculate the gravity wave potential energy density (GWPED) contained in the spectrometer temperatures. Results were separated into those with periods > 60 and <= 60 minutes. GWPED for waves with T < 60 min were in the range 7 - 15 J/kg, whereas those waves with T > 60 mins were in the range 10 - 150 J/kg. A relatively clear maximum in the GWPED for the former group occurred around January 27th, which is close to the time of a minor stratospheric warming event. The authors interpret this coincidence as possible evidence that these longer period waves originate at tropospheric altitudes. The cubic spine fitted to the two wave groups is of doubtful value.

Images from the FAIM camera were used to calculate wavelengths and propagation directions of the waves and ripples detected in the images. These were separated into those with lambda > 15 km and those <= 15 km. In the case of flight 1 (Kiruna – Alomar and back making a triangle) waves with lambda > 15 were either NW or SE, whereas those with lambda <= 15 km tended to be SW or NE. The highest occurrence rate of waves occurred in both legs when the plane was passing over the highest mountain peaks.

The manuscript is well organised and the data is clearly presented. The methods used

to identify the gravity waves in the image sequences are correct and the description of the methods used are clear. The text includes an appropriate set of references. The work is suitable for publication in ACP, provided that the minor points highlighted in the specific comments and in the technical corrections below are addressed.

Specific Comments On page13, the authors attempt to use the airglow brightness images to deduce something about whether static and dynamic instability is the dominant mechanism generating the ripples in flight 1 or flight 5. This is an interesting idea, but it is based entirely on assumptions which may or may not be true. In the absence of horizontal wind and temperatures data (see page 12, lines 31-33), we cannot say. This passage also assumes a relation between airglow brightness and temperature, which does not always hold strictly as pointed out in lines 17-19 on page 13.

A really useful reference on this point is the recent paper by Li et al. (2017), in which the authors study statistically the relation between ripples and the background atmosphere. Some of the statements made in the current manuscript are not supported in the work by Li et al. (2017), e.g., line 30 on page 11 states (referring to ripples) "They move with the background wind . . .". Li et al. (2017) report that less than half of the ripples examined moved with the background wind, and were in fact real wave structures that are difficult to distinguish from real instability features.

The authors should read Li et al. (2017) and revise the current manuscript in the light of the results presented there.

Technical corrections Page 1, line 27; "evolvement" –> "evolution". Page 1, line 29; "Special emphasize is put . . ." –> "Special emphasis is placed . . .". Page 2, line 29; insert "(BV)" after "Brunt-Väisälä". Page 3, line 18; insert "(GWPED)" after "density". Page 3, line 24; omit "of" in "on board of the DLR". Page 4, line 22; omit "certainly". Page 6, line 10; replace "looked up" by "found". Page 6, line 18; insert " for the calculation of N" after "necessary". Page 6, line 24; "catches" –> "includes". Page 7, line 8; replace "The used 2D FFT algorithm needs equidistant data." by "The 2D FFT algo-

rithm employed requires equidistant data.". Page 9, line 21; insert ";" after "also. Page 9, line 28; "cut" –> "reduced". Page 11, line 15; year of reference (2009) is inconsistent with line 7 on page 17. Page 12, line 17-18; suggest "Therefore, we conclude that gravity waves with periods longer than 60 min are more likely to could to a larger part be generated in the troposphere than gravity waves with periods shorter than 60 min." Page 14, line 7; "In the same time" –> "At the same time". Page 14, line 10; "could not be observed." –> "were not observed." Page 17, lines 5-7; year of reference (1995) is inconsistent with line 15 on page 11. Page 20, line 10; "preceding" –> "subsequent". Page 21, line 6; "stand for" –> represent. Page 24, Figure 6(a) and 6(b); dashed grey line is very faint. Use a darker colour. Page 26, Figure 8(a); the grey line that shows the orography is so faint that it is almost impossible to see it. Page 29, Figure 11(a); the grey line that shows the orography is so faint that it is almost impossible to see it.

References Li, J., T. Li, X. Dou, X. Fang, B. Cao, C.-Y. She, T. Nakamura, A. Manson, C. Meek, and D. Thorsen (2017), Characterstics of ripple structures revealed in OH airglow images. J. Geophys. Res. Space Physics, 122, 3748-3759, doi:10.1002/2016JA023538

---

## Referee Comment (RC3) · Anonymous Referee #3 · 17 Jan 2019

Review of "Observations of OH-airglow from ground, aircraft, and satellite: investigation of wave-like structures before a minor stratospheric warming" by Wüst et al. (acp-2018-2012)

The authors study pre-SSW gravity waves from airglow and/or temperature measurements in early 2016 using measurements of four different instruments: SABER-TIMED space radiometer, GRIPS9 (Kiruna) and GRIPS14 (Alomar) ground-based spectrometers, and FAIM imager (onboard FALCON aircraft). Following the work in Wüst et al. (2016), the authors derive time variation of BV frequency at the OH layer from SABER and, in combination with GRIPS temperatures, gravity wave potential energy density. They also derived short-time series of GW spectra and propagation direction, and their time variations from two FAIM flights (one of them right before a minor SSW). They detect highest GW occurrence over mountains. They also found dominance of small-scale GW contribution a couple of weeks before the SSW, which was not the case just before the SSW. Leaning on SABER-GRIPS BV frequency evolution and ECMWF data, they concluded that the small-scale waves in the first case were due to convective instability whereas they were due to dynamical instability in the second case. They also conclude that short period waves are generated in the higher stratosphere and above.

The paper is well written and organized, although some explanations could be simpler (particularly in the discussion). The English is ok.

I recommend the manuscript for publication, once the following suggestions and comments are taken into account.

General comments

The introduction does not include a description of previous results and the state-of-the-art in the field of GWs, in particular, before or during SSWs. Also, there is not a description of the scientific interest of the results presented here. Something similar happens in the discussion, which is not put into context of results from other authors or measurements. Indeed, there are previous publications (particularly regarding large scale features) that are not mentioned here.

The authors make use of measurements of several variables from 4 different instruments. In several places in the text, it is hard to know the instrument they are referring to or the calculations they are using. That makes the reading slow. For example, Sect. 4.2.1 shows calculations of GWPED that need from GRIPS temperature anomalies and periods, but these and their estimation are not shown nor even discussed anywhere. This happens more often (see comments below) and I recommend the authors reading the manuscript carefully with this criticism in mind in order to address this issue.

GRIPS14, observing over Alomar, is not used in the analysis. Only its 15-day mean temperatures and intensities are plotted but they are not further analyzed nor used for the discussion. Some information on wave propagation direction could be extracted when combining GRIPS9 (at Kiruna) with GRIPS14 (as in Wüst et al., 2018), perhaps also in the context of FAIM measurements. In any case, the results from GRIPS14 could support (or not) those from GRIPS9 and should be analyzed in parallel here. Additionally, they start early in January and can extend the time series longer.

In the discussion section, the author's conclusions are more a consistency with the behavior expected. For example, Flight 1 is just consistent with dynamical instability as the origin for ripples and Flight 5, right before the SSW, is not. This subtle difference is important because there is not an examination of other possible sources or very strong

evidence from these results behind that idea (on the one hand, it is based on the assumption that changes in brightness are only due to the generating GW; on the other hand, they only have two 2 days of measurements). That should be clear in the text.

Detailed comments

P2Sect.1. The introduction should be revised. The research is not put into context and the scientific scope of the paper needs to be better described. Just studying gravity waves is not an argument for a scientific paper. Please, include an explanation of the scientific interest.
P1. L18-24. Provide a small introduction of FAIM.
P1L20. Small-scales, write how small.
P1L21. Smaller aperture. How smaller?
P2.Sect.2: The instruments are poorly described. It is not easy to understand what and how exactly they measure.
Sect. 2.1: Unless you know GRIPS before reading this paper, it is not easy to know how exactly the instrument measures airglow. It is not even clear here that GRIPS is not an imager. What is the spectral resolution? Perhaps describing it here with more detail would help.
P3L7. Are these noise or systematic errors? Include a description of major sources of uncertainty.
P3L7. Include reference for temperature retrievals.
P3L11 Write observation angles for the 4 FoVs for GRIPS 9
P3L19. Shortly describe how you derive temperatures. Provide errors and error sources.
P3L25. Write the OH transitions this instrument is sensitive to.
P3L31. Please, indicate range.
P4L10. It is not clear. Are they analyzed or not?
P4L20. SABER is described in many papers. Better a reference to one of those than to a webpage that may eventually stop working.
P4L28. Remsberg et al. compared SABER v1.07 temperatures but you are using v2.0. Provide biases for v2.0, wether indicating v1.07-v2.0 comparisons or comparisons of v2.0 with other space and ground based instruments, which are already available.
P4L28-32. The authors are mixing here noise and systematic errors. Comparisons with other instruments should be commented in the context of systematic errors. SABER MLT temperature main errors are due to atomic oxygen uncertainties (Remsberg et al. 2008; Garcia-Comas et al. 2008). Also the biases strongly depend on latitude.
P4L32. For coherence, shortly comment on OH VER uncertainties.
P5L7. What do you mean by 500m negligible compared to 2000m FWHM? Please, quantify. Also note that SABER vertical sampling is several times smaller than its FOV.
P6L7. Insert 'Brünt-Vaisala (BV)'
P6L10. One really needs Wüst et al. 2016 in one hand when reading this manuscript, which is not useful. Please, shortly describe why shorter and longer than 60 min.
P6L23. For what transition?
P7L6. Could you better explain why airplane shaking prevents deriving period and phase speed? What is the error in the wavelength due to this shaking?
P7L8: Delete 'used'
P7L15: Please, clarify why you use here 87km and you mention 84km in previous section.
P7L15: Please, quantify the effects of layer altitude.
P7L19. Please, show in Fig. 1 the resulting image after applying this filter.
P8L4. According to what instrument?
P8L8. starts to rise by -> rises
P8L8. varies -> oscillates
P8L9. layer altitude
P8Sect.4.1. Fig. 2 is full of interesting things. I recommend including a more detail

description of the figure here.

P8L10. What SABER intensity is compared here? Averaged over the layer? Peak intensity? Does this choice make a difference?

P8L11. Only SABER and ALOMAR show a 4-6 day pronounced periodicity. GRIPS-9 periodicity is 9 days (one should not assume measurement for 15Feb is a maximum.

P8L12. Not in GRIPS 14.

P8L13. Include SABER OH*-temperatures. If comparable, that would somehow justify the use of SABER BV frequencies.

P8L18. Please, perform the same analysis with GRIPS 12 since it has a longer time coverage and also, if combined with GRIPS9, some information on horizontal propagation could be extracted.

P8L20. Describe here the temperature anomalies (amplitudes) you are using and how you estimated them.

P8L20. GRIPS temperature amplitudes

P8L21. There is no dashed line in Fig. 5

P8L22. Include 15-day averages in plot and discuss here in terms of fluctuations around the linear fit.

P8L24. Shortly describe criteria here.

P9L8 principal -> principle

P9L12. I guess that the authors mean an image horizontal coverage instead of spatial resolution, in contrast to the FoV used in this manuscript to refer to the spatial resolution for GRIPS. Please, homogenize definitions.

P8L14. ... and it also varies with OH layer altitude.

P9L15. What do you mean by time difference images? Explain how you treat several images overlapping.

P9L27. in sensitive -> is sensitive

P9L28. Why is the horizontal coverage cut to 26x26?

P9L30. What do you mean by 'small-scale' here?

P9L31. But the wavelengths smaller than 15km (1/k = [1/0.1,1/0.15]) appear very strongly at 17:40-17:55. Don't they?

P10L12-13. This info is not accurate, not used, not analyzed. The reader may loose attention to the central point of the FFT analysis.

P10L16. What do you mean by this? What do you think it is causing this large mean intensity?

P10L16. What do you mean by saying this? What do you think it is causing this large intensity? For previous flight, you just mentioned that mean intensity changed too much for long wavelenghts analysis....

P10L16 maximal -> maximum

P11Sect.5. The discussion gets complicated in some paragraphs. Please, re-read and simplify (this specially holds for reasoning in pages 12-13).

P11L3. A better description of the event, including dates of SSW onset and polar vortex displacement and recovery would be more useful.

P11L3. Delete 'the' before 'January'

P11L6-7. Include reference.

P11L12. Better than 'neglecting the effect of planetary waves' (which are the responsible for the polar vortex displacement mentioned above', you could write 'We expect the following effect on the zonal means.

P11L15. Mulligan et al. is missing in the reference list. Grygalashvyly (2015) and Garcia-Comas et al. (2017) should be included in this list.

P11L16. Explain why height and thickness are not anticorrelated in Fig 3.

P11L17. Insert 'According to SABER measurements,'

P11L18. also and particularly during February 2016 (see Fig. 2 and 3).

P12L3. vertical -> horizontal

P12L10-11. This may confuse the reader. Better saying "winds in the upper stratosphere

were stronger than in the upper troposphere"

P12L11. was -> were

P12L11. Easterly winds became weaker after Jan 23rd, which, for a continuous source of GWs, should have resulted in less overall filtering and more (E) GWs propagating to the mesosphere until Jan 28th. I can only glimpse the corresponding response in potential energy density for T<60min but the enhancement on the 27th is clear. Please, discuss on that. Perhaps, analysis of the next days in GRIPS9 time series (until Feb 2nd, as in Fig.4) could help. On the other hand, the change in FAIM total number of wave events before (Fig.9) and at the onset of the SSW (Fig. 12) does not clearly show any difference. Discuss on that also.

P12, L17. Insert 'according to GRIPS9 data,' after 'Therefore'

P12L18. This is too much of a conclusion based on zonal mean winds. Note the potential longitudinal variations or the length of the time series in Fig. 6.

P12L19 had the best chance -> had best chance

P12L21. Again, you should be careful when using zonal means from Fig. 14. I do not think you can resolve measurements over Kiruna using that information alone.

P12L25 Please, rewrite sentence

P12L26. I do not agree that the wind profile is rather flat before Jan 31st. There is a wind reversal around the stratopause and in the troposphere. What can be inferred from GRIPS14 measurements?

P12-13 The conclusions the authors reach are not put into context of results from other authors here, in particular, those regarding larger scale features (e.g.., Gerrard et al., 2011).

P13L2. Insert '(see Fig. 5)'

P13L3. What 'airglow brightness maps'?

P13L2. 'Since the measurements were taken in winter'

P13L6. What do you mean by 'overall' here? Note that you may eventually have inversion layers.

P13L8. Explain here what you define as the 'grey regions' of an airglow image

P13L7: Insert 'According to ECMWF data,'

P13L19. Please, make it clear that a correlation does not always hold (as in Pautet et al.) but, on average, a positive correlation between brightness and temperature should be a fair assumption, at least from mid-autumn to mid-winter. This was shown by WINDII and SATIs (Shepherd et al., 2006) but also by SABER, instrument that you use (Garcia-Comas et al., 2017).

P13L21. Why does the temperature gradient become zero? That depends on the amplitude of the wave. Better saying 'becomes maximum'.

P13L21. The use of 'steepest' here leads to misunderstanding. Better saying 'the minimum (or, since it is negative, maximum in absolute value) temperature gradient'

P13L23. 'compared to' -> 'depending on'

P13L25. Do you mean the 'zonal wind shear'

P13L27. Could you be more precise and describe the bright airglow areas you are referring to? Legs 4 and 5? How do you know these small-scale structures are only caused by a larger dynamical instability instead of any other cause, like location or just time variation?

P13L29. Better than 'then this means' use 'then this is consistent with'

P14L2. Although I agree that causes for ripples at the onset of a SSW are more likely due to changes in static instability, I do not think this conclusion can be inferred from these measurements. Again, it seems to me just a consistency (and not a conclusion) with a smaller dynamical instability. This is in part because your assumption that the large changes in brightness are only due to the generating GW is too strong, but also because of the lack of statistics (just 2 days). Additionally, these conclusions should be put in the context of previous results, which also should be referenced here.

P14L17. Insert 'combined with SABER data'

P14L19. 'below the tropospheric jet' -> 'in the troposphere'

Fig.2-caption. L5. SABER temperature?

Fig.2-caption: Write SABER channel.

Fig.2. Instead of the hard-to-follow description in the caption, just remove non-reliable data according.

Fig. 2. Please, change color code. It is not possible to differentiate most of them from others.

Fig. 4, L4. Indicate year of campaign.

Fig. 4. For coherence with panel a), include SABER temperatures in panel b) and discuss comparisons in text.

Fig. 5. The linear fit is not completely convincing. Indicate correlation and discuss in text.

Fig. 5-caption: Insert 'SABER' or 'derived from SABER'.

Fig. 6. Why do these data end on the 30th and not Feb. 2nd, as in Fig. 4?

Fig 6. L7. GRIPS 9

Figs. 9 and 12. I think that combining these two figures, that is, including the results for the two flights in the same plots would be interesting to see.

Fig.14. Lower panel is not needed for the discussion and does not provide additional useful information. Please, remove.

References

Gerrard et al., Observations of in-situ generated gravity waves during a STE event, Atmos. Chem. Phys., 11, 11913-11917, https://doi.org/10.5194/acp-11-11913-2011, 2011.

Grygalashvyly, M.: Several notes on the OH layer, Ann. Geophys., 33, 923–930, https://doi.org/10.5194/angeo-33-923-2015, 2015.

Garcia-Comas, M., et al.: Mesospheric OH layer altitude at midlatitudes: variability over the Sierra Nevada Observatory in Granada, Spain (37N, 3W), Ann. Geophys., 35, 1151–1164, https://doi.org/10.5194/angeo-35-1151-2017, 2017.

---

## Author Comment (AC3) · 29 Mar 2019

The comment was uploaded in the form of a supplement:
https://www.atmos-chem-phys-discuss.net/acp-2018-1012/acp-2018-1012-AC3-supplement.zip

---

## Author Response (AR1)

We would like to thank the anonymous referee for his valuable comments. We answered all of them and changed the manuscript accordingly. Please find the details below in orange.

General comments:

This manuscript describes and analyzes observations of OH airglow emissions from different platforms (particularly from the FALCON aircraft) during a field campaign in Scandinavia in January 2016. The paper is in general well written and easy to fol-

low. Although it does not provide any really new insights into the topic, the manuscript should in my opinion by published. It will probably be complemented by other publications dealing with the same field campaign. I recommend accepting the manuscript subject to minor revisions.      Below I offer some (mainly really minor) suggestions for improvements.

Specific comments:

Page 1, line 29: 'emphasize' -> 'emphasis' Done

Page 2, line 12: 'mounted at' -> 'mounted on' ? I think that's right, I changed it also in

section 2.2

Page 2, line 14: 'all other airborne measurements address heights of ca. 20 km and below.'

It's not entirely clear, what this part of the sentence refers to? To other instrumentation on Falcon? Yes, I added this information.

Page 3, line 25: I suggest replacing 'x' in '320 px x 256 px' by ' times' (I assume you use LaTex? ) I use Word, but I think I found the sign you meant in the formula editor, I substituted "x" when it was used in the sense of "times" in the whole manuscript

Page 3, line 30: '(compare Fig. 7 and 10).'

not sure, how this can be seen in Figures 7 and 10? Do the arrows indicate the flight track? Yes they do, I added this info in the caption.  The observed area is sometimes left, sometimes right of the arrows. Yes, we put the arrows where we thought there is enough space and where it fits best. But we can change it if you like.

Page 4, line 22: 'and upper levels'

It's not entirely clear, what 'upper levels' refers to. It may refer to 'upper atmospheric levels'or 'levels of excitation'. It's most likely the latter. Please specify. I mean higher altitude levels and added this information

Page 4, line 24: delete comma before '1.04 and 1.06' Done

[Figure]

Page 4, lines 28 – 30: Perhaps a paper on the validation of SABER temperature observations can be cited here? … Done (Dawkins, E. C. M., Feofilov, A., Rezac, L., Kutepov, A. A., Janches, D., Höffner, J., Chu. X., Lu, X., Mlynczak, M. G., and J. Russell III: Validation of SABER v2.0 operational temperature data with ground-based lidars in the mesosphere-lower thermosphere region (75–105 km). J. Geophys. Res.: Atmos., 123, 9916–9934. 10.1029/2018JD028742, 2018.)

Page 6, line 19: 'information are' -> 'information is' Done

Page 6, line 23: 'the height of 84 km'

Is this the height of maximum VER or a weighted, i.e. centroid altitude?
In this case it is the height of maximum VER, I added this info in the manuscript.

Figure 3: It would be good to mention explicitly in the Caption of Figure 3 that the year 2016 is shown. Done

Page 8, lines 24/25: 'information .. are' -> 'information .. is' Done

Page 9, line 7: 'flight legs parallel'

Perhaps add, e.g. 'roughly' or 'more or less', because the flight legs do not appear to be exactly parallel to the latitude/longitude circle? Done

Caption Figure 7: What exactly do you mean with 'Difference images'? This is later explained in the main text - as I found out - but perhaps it can be explained briefly in the caption, too? Done

Page 11, line 15: 'as well as height and intensity are anticorrelated.'

Regarding the anti-correlation between intensity and emission altitude the papers by Grygalashvyly (2014) and von Savigny (2015) may be cited, too. The first one provides a theoretical explanation for this anticorrelation and the second one shows the relationship for the OH(3-1) band (if I remember correctly), which is of importance to your work.

Grygalashvyly, M., G. R. Sonnemann, F.-J. Lübken, P. Hartogh, and U. Berger (2014), Hydroxyl layer: Mean state and trends at midlatitudes, J. Geophys. Res. Atmos., 119,

12,391–12,419, doi:10.1002/ 2014JD022094.

von Savigny, C., Variability of OH(3-1) emission altitude from 2003 to 2011: Long-term stability and universality of the emission rate - altitude relationship, J. Atmos. Sol.-Terr. Physics, 127, 2015.

Done

Page 13, line 19: 'So, if the vertical movements of atomic oxygen are due to a wave, one can conclude that the wave-induced vertical temperature gradient becomes zero where the brightness is maximal or minimal,'

I don't really understand this argument. Please explain.

I try it. Where the airglow brightness is maximal I should have maximal downward movement, where it is minimal I should have maximal upward movement.

When imagining gravity waves as only vertically oscillating coupled air parcels (and neglecting the horizontal component to make it easier), the vertical temperature profile will show a sine. The wave-induced temperature should be maximal where the air parcels are in their lowest position with respect to their rest position (they are deflected maximal downward from their rest position) and minimal when they are in their highest position with respect to their rest position (they are deflected maximal upward from their rest position). In the extreme points of the vertical profile of the temperature fluctuations, the vertical temperature gradient is zero and the brightness should be maximal or minimal.

Caption Figure 8: 'The horizontal line marks the wavelength of 15 km.' There are different horizontal lines. Please specify. I am sorry, the thicker lines moved through the image. I corrected the figure, now, only two lines are visible. This also holds for figure 11.
* * *
[Figure]

Atmos. Chem. Phys. Discuss.,
https://doi.org/10.5194/acp-2018-1012-RC2, 2019

[Figure]

We would like to thank the anonymous referee for his valuable comments. We answered all of them and changed the manuscript accordingly. Please find the details below in orange.

General Comments The authors present a selection of results from a campaign of observations on OH-airglow emissions recorded from the ground, and from an aircraft flown inside the Arctic Circle during January and February 2016. The ground-based observations were made using infrared spectrometers deployed at ALOMAR and Kiruna, while the aircraft measurements were made with a "Fast" airglow imager taken over

[Figure]

flight paths that included both observing stations.

These observations were supplemented with the inclusion of temperature and OH*
VER profiles from the TIMED-SABER satellite instrument near the time and location of

the observations, in addition to horizontal and meridional wind data from the ECMWF.

The SABER data have been used to calculate the mean height and thickness of the
OH* layer during the period of the observing campaign. There is good correspondence
between the variation of the OH* layer brightness measured by the ground based spec-
trometers and that obtained from the SABER VER measurements. The Brunt-Väisälä
(BV) frequency during the observing campaign was calculated for the OH* layer by
weighting it by the VER measurements. It showed a steady decrease throughout the
observing period – which was interpreted to imply a reduction in the static stability of
the atmosphere during that tim interval.

Combining the OH*-layer averaged temperature data from the infrared spectrometers
with the SABER temperature profiles, enabled the authors to calculate the gravity wave
potential energy density (GWPED) contained in the spectrometer temperatures. Re-
sults were separated into those with periods > 60 and <= 60 minutes.  GWPED for
waves with T < 60 min were in the range 7 - 15 J/kg, whereas those waves with T >
60 mins were in the range 10 - 150 J/kg.  A relatively clear maximum in the GWPED
for  the former group occurred around January 27th,  which is close to the time of a
minor stratospheric warming event. The authors interpret this coincidence as possible
evidence that these longer period waves originate at tropospheric altitudes. The cubic
spine fitted to the two wave groups is of doubtful value.

Images from the FAIM camera were used to calculate wavelengths and propagation
directions of the waves and ripples detected in the images. These were separated into
those with lambda > 15 km and those <= 15 km. In the case of flight 1 (Kiruna – Alomar
and back making a triangle) waves with lambda > 15 were either NW or SE, whereas
those with lambda <= 15 km tended to be SW or NE. The highest occurrence rate of
waves occurred in both legs when the plane was passing over the highest mountain
peaks.

The manuscript is well organised and the data is clearly presented. The methods used

to identify the gravity waves in the image sequences are correct and the description of the methods used are clear. The text includes an appropriate set of references. The work is suitable for publication in ACP, provided that the minor points highlighted in the specific comments and in the technical corrections below are addressed.

Specific Comments

On page 13, the authors attempt to use the airglow brightness images to deduce something about whether static and dynamic instability is the dominant mechanism generating the ripples in flight 1 or flight 5. This is an interesting idea, but it is based entirely on assumptions which may or may not be true. In the absence of horizontal wind and temperatures data (see page 12, lines 31-33), we cannot say. This passage also assumes a relation between airglow brightness and temperature, which does not always hold strictly as pointed out in lines 17-19 on page 13. Yes, that's true but as reviewer 3 pointed out, at least during this time of the year this assumption holds on average (Garcia-Comas et al., 2017; Shepherd et al., 2006). I integrated this information in the manuscript. I also pointed out the assumptions and used wind measurements at ALOMAR (Stober et al. 2017) in order to motivate at least the argumentation.

A really useful reference on this point is the recent paper by Li et al. (2017), in which the authors study statistically the relation between ripples and the background atmosphere. Some of the statements made in the current manuscript are not supported in the work by Li et al. (2017), e.g., line 30 on page 11 states (referring to ripples) "They move with the background wind *. . .*". Li et al. (2017) report that less than half of the ripples examined moved with the background wind, and were in fact real wave structures that are difficult to distinguish from real instability features.

The authors should read Li et al. (2017) and revise the current manuscript in the light of the results presented there.

Thank you very much for bringing this paper to my mind. I revised abstract, discussion

section and summary accordingly

Technical corrections

It seems to me that some of the technical corrections refer to the manuscript before it was improved during the quick review process. In this case I only mention "already corrected" as answer to the comment.

Page 1, line 27; "evolvement" –> "evolution". Already corrected

Page 1, line 29; "Special emphasize is put . . ." –> "Special emphasis is placed . . .". Done

Page 2, line 29; insert "(BV)" after "Brunt-Väisälä". Done

Page 3, line 18; insert "(GWPED)" after "density". Done

Page 3, line 24; omit "of" in "on board of the DLR". Done

Page 4, line 22; omit "certainly". Done, additionally I think it must be "are" instead of "is" just before the (now-deleted) "certainly"

Page 6, line 10; replace "looked up" by "found". Done

Page 6, line 18; insert " for the calculation of N" after "necessary". Done

Page 6, line 24; "catches" –> "includes". Done

Page 7, line 8; replace "The used 2D FFT algorithm needs equidistant data." by "The 2D FFT algorithm employed requires equidistant data.". Done

Page 9, line 21; insert ";" after "also. Already corrected

Page 9, line 28; "cut" –> "reduced". Done

Page 11, line 15; year of reference (2009) is inconsistent with line 7 on page 17. Already corrected

Page 12, line 17-18; suggest "Therefore, we conclude that gravity waves with periods longer than 60 min are more likely to could to a larger part be generated in the

troposphere than gravity waves with periods shorter than 60 min." Done

Page 14, line 7; "In the same time" –> "At the same time". Already corrected

Page 14, line 10; "could not   be observed." –> "were not observed." Done

Page 17, lines 5-7; year of reference (1995) is inconsistent with line 15 on page 11. Already corrected

Page 20, line 10; "preceding" –> "subsequent". Already corrected

Page 21, line 6; "stand for" –> represent. Already corrected

Page 24, Figure 6(a) and 6(b); dashed grey  line is very faint.  Use a darker colour. Already corrected during the quick review. Is it still too faint?

Page 26, Figure 8(a); the grey line that shows   the orography is so faint that it is almost impossible to see it.  Already corrected during the quick review. Is it still too faint?

Page 29, Figure 11(a);    the grey line that shows the orography is so faint that it is almost impossible to see it. Already corrected during the quick review. Is it still too faint?

References Li, J., T. Li, X. Dou, X.  Fang,  B.  Cao,  C.-Y.  She,  T.  Nakamura,  A. Manson, C. Meek, and D. Thorsen  (2017),  Characterstics  of  ripple  structures  re- vealed  in  OH  airglow  images. J. Geophys. Res. Space Physics, 122, 3748-3759, doi:10.1002/2016JA023538

[Figure]

Review of "Observations of OH-airglow from ground, aircraft, and satellite: investigation of wave-like structures before a minor stratospheric warming" by Wüst et al. (acp-2018-2012)

We would like to thank the anonymous referee for his valuable comments. We answered all of them and changed the manuscript accordingly. Please find the details below in orange.
Since answers to some comments (e.g. referring to the shaking of the airplane, the range of the different angles, the difference images, …) might be easier to follow we additionally deliver two videos, one for each flight as supplemental material. The left part shows the original image, the mid part shows the flight route colour-coded is the intensity averaged over the respective picture, and the right part shows the difference images.

The authors study pre-SSW gravity waves from airglow and/or temperature measurements in early 2016 using measurements of four different instruments: SABER-TIMED space radiometer, GRIPS9 (Kiruna) and GRIPS14 (Alomar) ground-based spectrometers, and FAIM imager (onboard FALCON aircraft). Following the work in Wüst et al. (2016), the authors derive time variation of BV frequency at the OH layer from SABER and, in combination with GRIPS temperatures, gravity wave potential energy density. They also derived short-time series of GW spectra and propagation direction, and their time variations from two FAIM flights (one of them right before a minor SSW). They detect highest GW occurrence over mountains. They also found dominance of small-scale GW contribution a couple of weeks before the SSW, which was not the case just before the SSW. Leaning on SABER-GRIPS BV frequency evolution and ECMWF data, they concluded that the small-scale waves in the first case were due to convective instability whereas they were due to dynamical instability in the second case. They also conclude that short period waves are generated in the higher stratosphere and above.

The paper is well written and organized, although some explanations could be simpler (particularly in the discussion). The English is ok.

I recommend the manuscript for publication, once the following suggestions and comments are taken into account.

General comments

The introduction does not include a description of previous results and the state-of-the-art in the field of GWs, in particular, before or during SSWs. Also, there is not a description of the scientific interest of the results presented here. Something similar happens in the discussion, which is not put into context of results from other authors or measurements. Indeed, there are previous publications (particularly regarding large scale features) that are not mentioned here. We included references in the manuscript. Since the focus of this paper is on smaller-scale features, we choose the references accordingly. We also included a short description of the scientific interest.

The authors make use of measurements of several variables from 4 different instruments. In several places in the text, it is hard to know the instrument they are referring to or the calculations they are using. That makes the reading slow. For example, Sect. 4.2.1 shows calculations of GWPED that need from GRIPS temperature anomalies and periods, but these and their estimation are not shown nor even discussed anywhere. This happens more often (see comments below) and I recommend the authors reading the manuscript carefully with this criticism in mind in order to address this issue.
We tried to make clearer which results are based on which instrument and also included a description of the GWPED derivation in the analysis section. In order not to lose focus and since the algorithm was published and discussed in detail in Wüst et al. (2016), we kept the description short.

GRIPS14, observing over Alomar, is not used in the analysis. Only its 15-day mean temperatures and intensities are plotted but they are not further analyzed nor used for the discussion. Some information on wave propagation direction could be extracted when combining GRIPS9 (at Kiruna) with GRIPS14 (as in Wüst et al., 2018), perhaps also in the context of FAIM measurements. In any case, the results from GRIPS14 could support (or not) those from GRIPS9 and should be analyzed in parallel here. Additionally, they start early in January and can extend the time series longer.

GRIPS 14 at ALOMAR is not used for the derivation of GW information since the weather situation at ALOMAR was not suitable during the time period when GRIPS 9 measured at Kiruna. This was mentioned in section 2.1 but we now additionally included this info in section 4.2.1 where the GWPED from Kiruna is shown.

In the discussion section, the author's conclusions are more a consistency with the behavior expected. For example, Flight 1 is just consistent with dynamical instability as the origin for ripples and Flight 5, right before the SSW, is not. This subtle difference is important because there is not an examination of other possible sources or very strong evidence from these results behind that idea (on the one hand, it is based on the assumption that changes in brightness are only due to the generating GW; on the other hand, they only have two 2 days of measurements). That should be clear in the text.
We tried to make it clear.

Detailed comments

P2Sect.1. The introduction should be revised. The research is not put into context and the scientific scope of the paper needs to be better described. Just studying gravity waves is not an argument for a scientific paper. Please, include an explanation of the scientific interest. Done
P1. L18-24. Provide a small introduction of FAIM. I assume it's page 2 not page 1. Inserted one sentence.
P1L20. Small-scales, write how small. Done
P1L21. Smaller aperture. How smaller? Done
P2.Sect.2: The instruments are poorly described. It is not easy to understand what and how exactly they measure. Since the focus of this publication is not the instruments, which are described in detail in separate publications, I extended these subsections to some extent (GRIPS more, FAIM less).
Sect. 2.1: Unless you know GRIPS before reading this paper, it is not easy to know how exactly the instrument measures airglow. It is not even clear here that GRIPS is not an imager. What is the spectral resolution? Perhaps describing it here with more detail would help. Done.
P3L7. Are these noise or systematic errors? Include a description of major sources of uncertainty. Done
P3L7. Include reference for temperature retrievals. Done
P3L11 Write observation angles for the 4 FoVs for GRIPS 9 Done
P3L19. Shortly describe how you derive temperatures. Provide errors and error sources. Done.
P3L25. Write the OH transitions this instrument is sensitive to. Done
P3L31. Please, indicate range. Done, yaw angle removed since it only changes the orientation of the FoV.
P4L10. It is not clear. Are they analyzed or not? No, they are not, I changed the sentence to "Therefore, these measurements are not part of this publication." too make it clear.
P4L20. SABER is described in many papers. Better a reference to one of those than to a webpage that may eventually stop working. Sentence deleted
P4L28. Remsberg et al. compared SABER v1.07 temperatures but you are using v2.0. Provide biases for v2.0, wether indicating v1.07-v2.0 comparisons or comparisons of v2.0 with other space and ground based instruments, which are already available. Done
P4L28-32. The authors are mixing here noise and systematic errors. Comparisons with other instruments should be commented in the context of systematic errors. SABER MLT temperature main errors are due to atomic oxygen uncertainties (Remsberg et al. 2008; Garcia-

Comas et al. 2008). Also the biases strongly depend on latitude. Information concerning the quality of v1.07 deleted and replaced with information concerning v2.0.

P4L32. For coherence, shortly comment on OH VER uncertainties. I found detailed info about the different temperature errors, but I found no publication where this info is provided for the VER.

P5L7. What do you mean by 500m negligible compared to 2000m FWHM? Please, quantify. Done Also note that SABER vertical sampling is several times smaller than its FOV. Done

P6L7. Insert 'Brünt-Vaisala (BV)' Already introduced on page 5

P6L10. One really needs Wüst et al. 2016 in one hand when reading this manuscript, which is not useful. Please, shortly describe why shorter and longer than 60 min. Done

P6L23. For what transition? Done

P7L6. Could you better explain why airplane shaking prevents deriving period and phase speed? What is the error in the wavelength due to this shaking?
The shaking translates the FoV by several pixels in a quasi-periodic manner and applies a motion blur on the images. The translation does not allow deriving the change of phase from consecutive images, but this information is crucial for calculating phase speed and period of the waves. The translation affects the whole image and therefore all wave crests within the image, the wavelength which is derived for each image individually is not influenced. The motion blur does not change the position of the wave crests, but it reduces the amplitude of the waves. The amplitude, however, is not used here. Info added in the text.

P7L8: Delete 'used' Done

P7L15: Please, clarify why you use here 87km and you mention 84km in previous section. We took 87 km since this is the "standard height" for the OH-airglow layer. A publication which is often cited here is Baker and Stair (1988). We clarified in the previous section that the value 84 km holds only for the time period analysed in Wüst et al., 2016 and that other values are possible. Furthermore, 84 km is the height of maximum VER, the centroid height is mostly slightly higher.

P7L15: Please, quantify the effects of layer altitude. +/-5 km in the altitude layer corresponds to +/-6% in the resolution and therefore also in the wavelength (calculated for a zenith angle of 5°), info added.

P7L19. Please, show in Fig. 1 the resulting image after applying this filter. Done, we additionally show a second example. Here it becomes clear why we choose a square of 26 km x 26 km for the analysis.

P8L4. According to what instrument? SABER, info added

P8L8. starts to rise by -> rises Done

P8L8. varies -> oscillates Done

P8L9. layer altitude Done

P8Sect.4.1. Fig. 2 is full of interesting things. I recommend including a more detail description of the figure here. Taking into account your comment on this figure at the end of the manuscript (Fig. 2. Please, change color code. It is not possible to differentiate most of them from others), I tried different versions but in every case the figure is either not readable for people who are "red-green blind" or the figure becomes confusing (when using different line styles, for example). The main purpose of this figure is to show the behavior of winter 2015/16 compared to the mean over all years. I agree with you that there are certainly many interesting things to deduce concerning the other years, however, I would like to keep the paper focused. Therefore, I decided to delete all curves but the mean and the one referring to winter 2015/16.

P8L10. What SABER intensity is compared here? Averaged over the layer? Peak intensity? Does this choice make a difference? It's the peak intensity (added info in the manuscript) and the choice makes no difference concerning the variations. I calculated the integrated intensities and compared them to the peak intensities. They correlate linearly with an $R^2$ of about 87%. Info added in the manuscript.

P8L11. Only SABER and ALOMAR show a 4-6 day pronounced periodicity. GRIPS-9 periodicity is 9 days (one should not assume measurement for 15Feb is a maximum. Changed to "In particular, they show pronounced periodicities in the range of some days". In order to avoid misunderstandings, I would like to mention that figure 4 does not include February, 15th, it ends at the beginning of February. I added this information in the figure caption to make it clear.

P8L12. Not in GRIPS 14. Here, we disagree. Could please have a second look at the figure taking into account the information about the x-axis I gave one comment above?

P8L13. Include SABER OH*-temperatures. If comparable, that would somehow justify the use of SABER BV frequencies. Done. By preparing the new figure, we realized two things: a) we originally used SABER data around Kiruna and not around Alomar; that is a contradiction to section 2.3 where we say that we use only SABER data around Alomar. b) There was a slight offset in the relative GRIPS intensities for Alomar. We corrected both.

P8L18. Please, perform the same analysis with GRIPS 12 since it has a longer time coverage and also, if combined with GRIPS9, some information on horizontal propagation could be extracted. Sorry, but I don't know which analysis you mean. You probably refer the GWPED. As mentioned above the weather at ALOMAR was bad when it was good at Kiruna (bad and good with respect to the derivation of GWPED)

P8L20. Describe here the temperature anomalies (amplitudes) you are using and how you estimated them. I think this information should be mentioned in section 3.1 where the analysis is described.

P8L20. GRIPS temperature amplitudes Can you please concretize your comment?

P8L21. There is no dashed line in Fig. 5 Sorry, it's solid.

P8L22. Include 15-day averages in plot and discuss here in terms of fluctuations around the linear fit. Can you please concretize why you would like to see a 15-day average here and a discussion of the fluctuations around the linear fit? With this figure I intend to show that the BV-frequency decreases overall even though it shows superimposed fluctuations. For the GWPED density I use the exact BV-values and not the linearly fitted ones. I would like to avoid confusing the reader with too many details which are not really necessary for the publication.

P8L24. Shortly describe criteria here. Done

P9L8 principal --> principle Corrected

P9L12. I guess that the authors mean an image horizontal coverage instead of spatial resolution, in contrast to the FoV used in this manuscript to refer to the spatial resolution for GRIPS. Please, homogeneize definitions. Sorry, but I do not understand this comment. We don't say anything about the spatial resolution in this line. Furthermore, this section refers to FAIM as mentioned at the beginning of the section.

P8L14. ... and it also varies with OH layer altitude. You probably mean page 9. Yes, that's true, but the change of the roll (in the following figure denoted as zenith angle) and pitch

[Figure]

angles dominates possible variations in the OH-layer height during one flight. The roll angle is 25° at maximum.

P9L15. What do you mean by time difference images? Explain how you treat several images overlapping. A difference image is derived by subtracting the intensity measured by each pixel from the intensity measured 10 s later by the same pixel. For this method, it is not a problem, if several images overlap.

P9L27. in sensitive --> is sensitive Corrected

P9L28. Why is the horizontal coverage cut to 26x26? This is the largest square size which does not contain any pixels outside the un-warped image region (marked in Fig. 1). This information is already given in section 3.2, therefore, I don't insert it here.

P9L30. What do you mean by 'small-scale' here? Wavelengths in the range of 15 km and less, info added in the text.

P9L31. But the wavelengths smaller than 15km (1/k = [1/0.1,1/0.15]) appear very

strongly at 17:40-17:55. Don't they? Yes they do, that's why we mentioned the time period 17:30–18:00 in the text.

P10L12-13. This info is not accurate, not used, not analyzed. The reader may loose attention to the central point of the FFT analysis. Deleted

P10L16. What do you mean by this? What do you think it is causing this large mean intensity? See answer to next comment

P10L16. What do you mean by saying this? This is just a result. We interpret it later in section 5. What do you think it is causing this large intensity? That's a good question which we cannot answer based on our measurements. The maximum is comparable in its height to the one in leg 2. The horizontal distance between both maxima does not contradict the assumption of gravity waves (which we make in section 5). For previous flight, you just mentioned that mean intensity changed too much for long wavelenghts analysis....

For the previous flight, we mentioned "The airglow brightness averaged over each picture shows local maxima during these three time periods (Fig. 8a)." The time periods we here refer to are the ones with "high Fourier amplitudes also in the range of small-scale features".

P10L16 maximal --> maximum In this line there exists two times the word "maximal" and after consultation of a dictionary I think one can use this word in both cases.

P11Sect.5. The discussion gets complicated in some paragraphs. Please, re-read and simplify (this specially holds for reasoning in pages 12-13). I re-arranged this section and hope that it became clearer.

P11L3. A better description of the event, including dates of SSW onset and polar vortex displacement and recovery would be more useful. Done

P11L3. Delete 'the' before 'January' Corrected

P11L6-7. Include reference. Done

P11L12. Better than 'neglecting the effect of planetary waves' (which are the responsible for the polar vortex displacement mentioned above', you could write 'We expect the following effect on the zonal means. Corrected

P11L15. Mulligan et al. is missing in the reference list. Already corrected in a former version Grygalashvyly (2015) and Garcia-Comas et al. (2017) should be included in this list. Done

P11L16. Explain why height and thickness are not anticorrelated in Fig 3.

The FWHM was calculated straight forward: the VER-maximum is searched between 70 km and 100 km. 50% of this maximum is calculated. Then, the maximal and minimal height, where the VER is greater than 50% of the VER-maximum is derived. The minimal height is subtracted from the maximal height and this value is denoted as the FWHM. I checked the profiles and the values make sense.

The VER-profile is to some part influenced by oscillations. Depending on the strength of the oscillations, this also influences the FWHM. The anti-correlation reported in literature is observed in a statistical sense. These might be two reasons why figure 3 looks different than expected. However, from the 15-day average shown figure 2 it becomes clear that overall the behavior of the OH*-height and the FHWM is not so far away from what we expect.

P11L17. Insert 'According to SABER measurements,' Done

P11L18. also and particularly during February 2016 (see Fig. 2 and 3). Information added

P12L3. vertical --> horizontal I reformulated this sentence in order to avoid confusion.

P12L10-11. This may confuse the reader. Better saying "winds in the upper stratosphere were stronger than in the upper troposphere" Corrected

P12L11. was --> were Corrected

P12L11. Easterly winds became weaker after Jan 23rd, which, for a continuous source of GWs, should have resulted in less overall filtering and more (E) GWs propagating to the mesosphere until Jan 28th. Yes that's true and agrees with our argumentation. We included the info about the date of wind speed weakening.

I can only glimpse the corresponding response in potential energy density for T<60min but the enhancement on the 27th is clear. Please, discuss on that.

Do you really mean T<60 min? I see the enhancement only for T>60 min. For T<60 min we can speculate about a maximum at Jan. 26th at least compared to mid January (18th) and end of January (28th). I included this info.

Perhaps, analysis of the next days in GRIPS9 time series (until Feb 2nd, as in Fig.4) could

help.

For figure 4, nightly mean temperatures are used. The quality criteria for deriving GWPED are higher. Unfortunately, the data quality of the ALOMAR measurements was relatively bad at the end of January, so further GPWED values cannot be derived.

On the other hand, the change in FAIM total number of wave events before (Fig.9) and at the onset of the SSW (Fig. 12) does not clearly show any difference. Discuss on that also.

The number of wave events changes: it becomes larger by a factor of ca. 1.5–2.0 (for wavelengths shorter or longer than 15 km). We included this information in fig. 9 and 12 as well as in the results and discussion section.

P12, L17. Insert 'according to GRIPS9 data,' after 'Therefore' Done

P12L18. This is too much of a conclusion based on zonal mean winds. Note the potential longitudinal variations or the length of the time series in Fig. 6. That's a misunderstanding; figure 14 does not show zonal means it depicts the wind profile for the grid point next to Kiruna. We additionally inserted a discussion of the meridional component:

"The meridional wind component evolves differently (Fig. 14 b) compared to the zonal one: the direction of the meridional wind varies over the whole height range between January 20th and 27th, 2019. Afterwards, this is not the case any more. If gravity wave filtering is driven by the meridional wind, one expects also in this case that the activity of gravity waves generated in the troposphere increases at the end of January."

P12L19 had the best chance --> had best chance Corrected

P12L21. Again, you should be careful when using zonal means from Fig. 14. I do not think you can resolve measurements over Kiruna using that information alone. That's a misunderstanding, figure 14 does not show zonal means, it shows the wind profile for the grid point next to Kiruna.

P12L25 Please, rewrite sentence Reformulated by taking also into account the next comment.

P12L26. I do not agree that the wind profile is rather flat before Jan 31st. There is a wind reversal around the stratopause and in the troposphere. Reformulated

What can be inferred from GRIPS14 measurements?

As mentioned above, GRIPS 14 measurements at ALOMAR are of insufficient quality for the calculation of GWPED (bad weather).

P12--13 The conclusions the authors reach are not put into context of results from other authors here, in particular, those regarding larger scale features (e.g.., Gerrard et al., 2011). Done

P13L2. Insert '(see Fig. 5)' Done

P13L3. What 'airglow brightness maps'? The use of the word "maps" is irritating, we mean information about the airglow brightness → corrected

P13L2. 'Since the measurements were taken in winter' I assume that you refer to line 5. There, I corrected it but in line 2 I don't know what to change.

P13L6. What do you mean by 'overall' here? Note that you may eventually have inversion layers. I changed the sentence to "Thus, as long as inversion layers do not exist, the vertical background temperature gradient is negative in the height range of the OH* layer. Static instability is therefore possible and independent of the existence of gravity waves."

P13L8. Explain here what you define as the 'grey regions' of an airglow image Done

P13L7: Insert 'According to ECMWF data,' Do you really mean line 7? This line does not refer to ECMWF data

P13L19. Please, make it clear that a correlation does not always hold (as in Pautet et al.) but, on average, a positive correlation between brightness and temperature should be a fair assumption, at least from mid--autumn to mid--winter. This was shown by WINDII and SATIs (Shepherd et al., 2006) but also by SABER, instrument that you use (Garcia--Comas et al., 2017). Thank you for this hint. We included it.

P13L21. Why does the temperature gradient become zero? That depends on the amplitude of the wave. Better saying 'becomes maximum'. In this part, we speak about the wave-induced temperature deviations from the atmospheric background, so we neglect the background. In regions of maximal or minimal wave-induced temperature, the air parcels are deflected maximal from their original position (rest position), so the vertical transport should be maximal there. In regions of maximal or minimal temperature, the temperature gradient is zero.

P13L21. The use of 'steepest' here leads to misunderstanding. Better saying 'the minimum (or, since it is negative, maximum in absolute value) temperature gradient' See answer to comment above, in the regions where the wave has its zero-crossing, the temperature does not change and the gradient is steepest.

P13L23. 'compared to' --> 'depending on' Sentence deleted, it probably causes more confusion than it helps.

P13L25. Do you mean the 'zonal wind shear' yes and info added

P13L27. Could you be more precise and describe the bright airglow areas you are referring to? Legs 4 and 5? Done

How do you know these small-scale structures are only caused by a larger dynamical instability instead of any other cause, like location or just time variation? We think that these small-scale features have wave-like structure. This is due to the use of the 2D FFT. As reviewer 2 pointed out Li et al., (2017) showed that wavelengths in the range of ripples do not necessarily have to be instability features, they can also be secondarily generated small-scale gravity waves. I included this hint and changed the discussion but also abstract and summary accordingly.

P13L29. Better than 'then this means' use 'then this is consistent with' Done

P14L2. Although I agree that causes for ripples at the onset of a SSW are more likely due to changes in static instability, I do not think this conclusion can be inferred from these measurements. Again, it seems to me just a consistency (and not a conclusion) with a smaller dynamical instability. This is in part because your assumption that the large changes in brightness are only due to the generating GW is too strong, but also because of the lack of statistics (just 2 days). Sentence reformulated and message weakened.

Additionally, these conclusions should be put in the context of previous results, which also should be referenced here. Most of the manuscripts concerning ripples only treat single events. The manuscript which is probably based on the largest data set is Li et al. (2017), which I therefore mention it here.

P14L17. Insert 'combined with SABER data' Done

P14L19. 'below the tropospheric jet' --> 'in the troposphere' Done

Fig.2--caption. L5. SABER temperature? VER, thank you

Fig.2--caption: Write SABER channel. Done

Fig.2. Instead of the hard--to--follow description in the caption, just remove non--reliable data according. Number of lines reduced, see answer to comment P8 Sect.4.1. Info now not necessary any more.

Fig. 2. Please, change color code. It is not possible to differentiate most of them from others. Number of lines reduced, see answer to comment P8 Sect.4.1.

Fig. 4, L4. Indicate year of campaign. Done

Fig. 4. For coherence with panel a), include SABER temperatures in panel Done b) and discuss comparisons in text. Done

Fig. 5. The linear fit is not completely convincing. Indicate correlation and discuss in text. The linear fit can't be convincing since the temperature development, which is not linear either, must also be "visible" in the development of the BV frequency. The linear fit is only to guide the eye. I adapted the figure caption and the description of the figure in section 4.1 ("Between day 1 and 60 of 2016, the OH*-equivalent (angular) BV frequency decreases overall. If one approximates the OH*-equivalent (angular) BV frequency linearly, the approximated values range from ca. 0.022 1/s to 0.020 1/s (Fig. 5, solid line). However, superimposed fluctuations are visible, which reach ca. 13% deviation from the linear fit at maximum.")

Fig. 5--caption: Insert 'SABER' or 'derived from SABER'. Done

Fig. 6. Why do these data end on the 30th and not Feb. 2nd, as in Fig. 4? In figure 4, nightly mean values are plotted. For the derivation of the GWPED stricter quality criteria apply and the nights in February didn't meet them. Fig 6. L7. GRIPS 9 Done

Figs. 9 and 12. I think that combining these two figures, that is, including the results for the two flights in the same plots would be interesting to see. Done

Fig.14. Lower panel is not needed for the discussion and does not provide additional useful information. Please, remove. But we use this panel, page 14 ll. 29&30 (version with accepted changes).

References

Gerrard et al., Observations of in-situ generated gravity waves during a STE event, Atmos. Chem. Phys., 11, 11913-11917, https://doi.org/10.5194/acp-11-11913-2011, 2011.

[revised manuscript text omitted]